# Human-MME: A Holistic Evaluation Benchmark for Human-Centric Multimodal Large Language Models

**Yuansen Liu**[1][*]  **Haiming Tang**[1][*]  **Jinlong Peng**[2][*]  **Jiangning Zhang**[2]  **Xiaozhong Ji**[3]
**Qingdong He**[2]  **Donghao Luo**[2]  **Zhenye Gan**[2]  **Junwei Zhu**[2]  **Yunhang Shen**[2]
**Chaoyou Fu**[3]  **Chengjie Wang**[2]  **Xiaobin Hu**[1]✉  **Shuicheng Yan**[1]

[1]National University of Singapore    [2]Tencent Youtu Lab    [3]Nanjing University

## ABSTRACT

Multimodal Large Language Models (MLLMs) have demonstrated significant advances in visual understanding tasks. However, their capacity to comprehend human-centric scenes has rarely been explored, primarily due to the absence of comprehensive evaluation benchmarks that take into account both the human-oriented granular level and higher-dimensional causal reasoning ability. Such high-quality evaluation benchmarks face tough obstacles, given the physical complexity of the human body and the difficulty of annotating granular structures. In this paper, we propose Human-MME, a rigorously curated benchmark designed to provide a more holistic evaluation of MLLMs in human-centric scene understanding. Compared with other existing benchmarks, our work provides three key features: **(1) Diversity in human scene**, spanning 4 primary visual domains with 15 secondary domains and 43 sub-fields to ensure broad scenario coverage. **(2) Progressive and diverse evaluation dimensions**, evaluating the human-based activities progressively from the human-oriented granular perception to the higher-dimensional multi-target and causal reasoning, consisting of eight dimensions with 19,945 real-world image question pairs and an evaluation suite. **(3) High-quality annotations with rich data paradigms**, constructing the automated annotation pipeline and human-annotation platform, supporting rigorous manual labeling by expert annotators to facilitate precise and reliable model assessment. Our benchmark extends the single-person and single-image understanding to the multi-person and multi-image mutual understanding by constructing the choice, short-answer, grounding, ranking and judgment question components, and complex question-answer pairs of their combination. The extensive experiments on 20 state-of-the-art MLLMs effectively expose the limitations and guide future MLLMs research toward better human-centric image understanding and reasoning. Data and code are available at https://github.com/Yuan-Hou/Human-MME.

## 1 Introduction

Recent advances in multimodal large language models (MLLMs) have demonstrated remarkable capabilities in perceptual understanding and reasoning for general comprehension tasks. Among various types of scene data, human-centric images represent a particularly critical domain due to their prevalence in real-world data (Wang et al., 2024; Gkioxari et al., 2018). Compared to general image understanding, human-centric image understanding imposes greater challenges on models. These tasks require not only fine-grained perception (*e.g.*, eyebrow, accessories) and recognition of physical complexity, but also highly sophisticated causal reasoning abilities (Xiao & Yamasaki, 2024; Yang et al., 2022). Consequently, a thorough evaluation of the capabilities and limitations of existing MLLMs within this domain is critical. Such an investigation is fundamental to progress in

---

[*]Yuansen Liu, Haiming Tang, and Jinlong Peng are co-first authors
✉ Xiaobin Hu is corresponding author

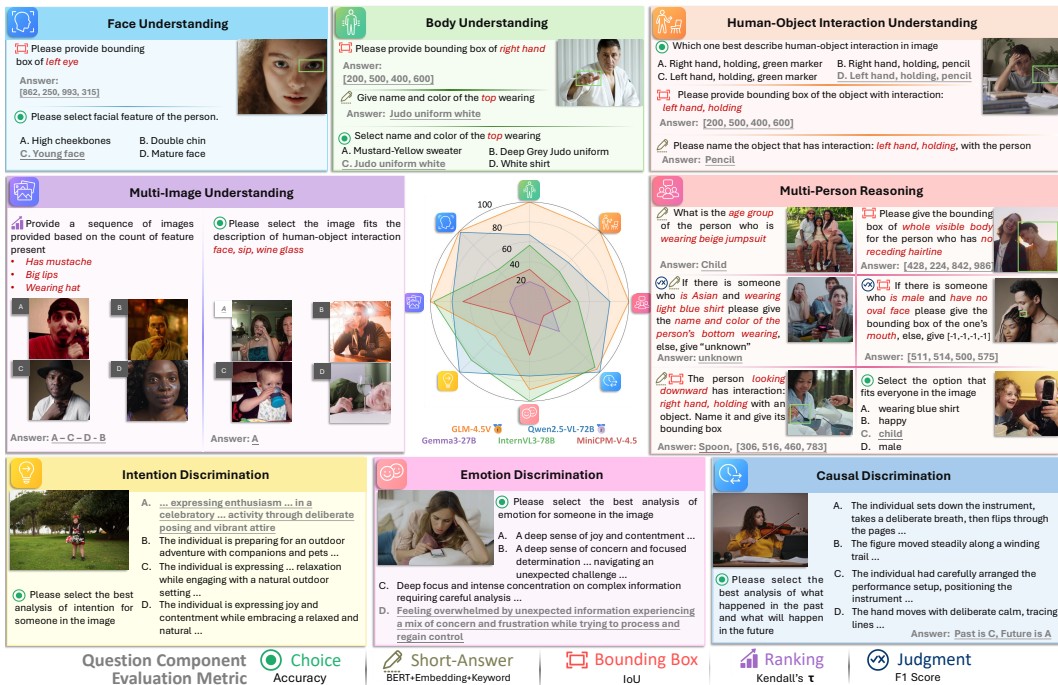

Figure 1: **Overview of Human-MME**: The progressive and diverse evaluation dimensions can be divided into eight aspects from the human-oriented granular dimension perception (*e.g.,* face, body, human-object interaction understanding) to higher-dimension reasoning (*e.g.,* multi-image and multi-person understanding, intention, emotion, cause discrimination).

both theoretical and the evolving human-oriented framework of MLLMs. However, existing benchmarks rarely attempt to explore the fine-grained human-centric image understanding, but predominantly focus on the general content comprehension. These benchmarks usually suffer from three key limitations in human-centric scenes: **(1)** Overly simplistic evaluation settings that inadequately represent the full spectrum of human-centric activities. **(2)** Lack of comprehensive dimensions to take into account both the granular level and higher-level spatial and reasoning perception. **(3)** Low annotation quality and limited question-answer paradigms to handle a broader spectrum of sophisticated and diverse reasoning tasks. Such deficiencies preclude a holistic evaluation of the MLLMs' inherent capacity for human-centric scene understanding.

To address these limitations, we propose Human-MME, the first comprehensive benchmark for evaluating MLLMs toward human-centric image scene understanding via progressive and diverse evaluation dimensions from granular dimension perception and higher-dimension reasoning, as shown in Figure 1. Compared to the existing benchmarks, our benchmark distinguishes itself through three key innovations: **(1) Diversity in human scene.** The benchmark consists of 43 distinct and fine-grained visual scenarios to support the comprehensive human scene perception. **(2) Progressive and diverse evaluation dimensions.** The evaluation is structured to progressively assess MLLMs' capabilities from granular human-oriented perception to complex spatial and causal reasoning, which is quantified across eight dimensions via a dataset of 19,945 real-world image-question pairs and a comprehensive evaluation suite. **(3) High-quality annotations with rich data paradigms.** Human-MME advances beyond single-image analysis to multi-image, multi-person mutual understanding and introduces a suite of tasks: choice, short-answer, ranking, grounding, identifying, and judgment. These high-quality tasks are facilitated by our annotation platform, which enables expert annotators to efficiently verify and correct automatically annotated labels through intuitive operations.

To assess Human-MME effectiveness, we perform a comprehensive benchmarking using 20 state-of-the-art open-source and proprietary MLLMs. Our benchmark reveals several interesting findings in Section 3.3. We hope that our benchmarking results, evaluation findings, and benchmark itself will inspire and guide future research toward developing more capable and robust human-oriented MLLMs. In summary, our key contributions are as follows:

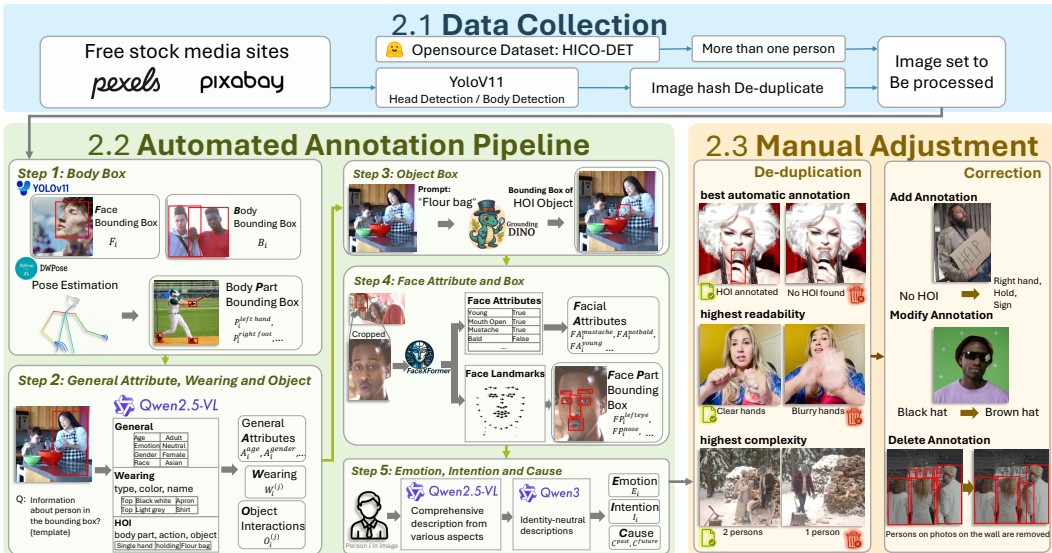

Figure 2: **Curation of Human-MME** consists of: **(1)** Data collection to provide images for annotation and QA generation (Section 2.1); **(2)** Automated annotation to provide the original feature set for each person $i$ in image (Section 2.2); **(3)** Manual adjustment to ensure annotation quality for final question-answer construction (Section 2.3); **(4)** Construction of question-answer pairs using the features extracted in the image (Section 2.4, not shown in the figure).

- We introduce Human-MME, a novel human-centric image benchmark for MLLMs that emphasizes progressive assessment across eight dimensions, from fine-grained human comprehension to higher-level intricate spatial and causal reasoning perception.

- We build up a user-friendly annotation pipeline and platform facilitating rich and high-quality data paradigms, covering 43 fine-grained visual scenarios and 19,945 real-world image-question pairs. The type of data paradigms breaks the restriction of single-image, single-person and single-question paradigms, and extends to multi-image mutual understanding and sequential complex question-answer pairs.

- To the best of our knowledge, we are the first to conduct a holistic evaluation of MLLMs on human-centric image perception in a progressive manner that considers both fine-grained perceptual dimensions and higher-level reasoning. Extensive experiments on 20 state-of-the-art MLLMs effectively expose current limitations and provide guidance for future models toward improved human-centric image understanding and reasoning. All data, the annotation platform, and the code will be publicly released to support reproducibility.

## 2 CURATING HUMAN-MME BENCHMARK

### 2.1 DATA COLLECTION

We collect a total of 54,122 images from Pexels and Pixabay. In addition, the HICO-DET (Chao et al., 2018) dataset, which contains a total of 47,776 images, has not been reported as training data in any of the models we evaluated, which makes it suitable for use in our benchmark.

For images from free media sites, we use pre-trained YOLOv11 (Jocher & Qiu, 2024) models to detect face and body bounding boxes and only retain the ones with detection results. To reduce redundancy, the extracted images are further de-duplicated using image hashing (Buchner, 2021). For the HICO-DET (Chao et al., 2018) dataset, we select images containing more than one person and compared the original annotations with our automatically generated ones for further filtering. After further annotation and de-duplication, we obtain 8,010 images from the free media sites and 8,755 images from the open-source dataset, with a total of 16,765 high-quality images.

## 2.2 Automated Annotation Pipeline

After collecting the image data, we complete the automated annotation of these images in five steps as shown in Figure 2. This automated annotation pipeline produces 13 different types of bounding boxes, 42 binary facial features, and four person-level attributes. It also enumerates eight categories of clothing worn by the person (covering details such as color, type, and name) and captures interactions between the person and objects (including the interacting body part, action, object name). In addition, it extracts three types of higher-dimensional features. The symbolic representation and interpretation of the extracted features can be found in Table 8. Here we explain step by step how each feature of person $i$ in the image is extracted. Detailed information about automated annotation pipeline can be found in the Appendix B.

**Face bounding box, body bounding box and body parts bounding boxes.** In step 1, we use a pre-trained YOLOv11 detector (Jocher & Qiu, 2024) to produce candidate body and face bounding boxes. In parallel, we apply DWPose (Yang et al., 2023) to each image to obtain whole-body pose estimates per person instance with 134 keypoints. After establishing the correspondence between pose estimation and face/body bounding boxes using geometric relationships, for each pose estimation, we align the body and face boxes and extract boxes for both hands and both feet.

**General attributes, wearing and human-object interactions (HOI).** In step 2, we use the matched bounding box obtained from step 1 to isolate each person instance within the image. For each instance, we query Qwen2.5-VL-72B (Bai et al., 2025b) following a JSON template to extract: (1) A set of factual attributes including age group, gender, race and emotion. (2) Each clothing item with their types, colors and names. (3) Interaction relationship with objects including: body parts that conduct the interaction, the verb phrases of the actions and names of the objects.

**Bounding boxes for HOI.** In step 3, the original image and the names of HOI objects are given to Grounding DINO (Liu et al., 2023), which predicts the bounding boxes for HOI objects. The bounding box of an object is also combined with the bounding box of body parts to refine the body part information of the HOI.

**Facial attributes and bounding boxes for facial parts.** In step 4, for each individual, the corresponding face region is passed to FaceXFormer (Narayan et al., 2024) for facial attributes and landmarks recognition. Facial attributes from FaceXFormer are 40 binary values, for example "Bags Under Eyes" and "Bangs", in the same format as CelebA (Liu et al., 2015). In addition, two binary values are extracted from the head pose prediction. Bounding boxes for facial parts including eyes, eyebrows, mouth and nose are extracted from the 68 landmarks predicted.

**Intention, Emotional analysis, cause (past) and consequence (future) narratives.** In step 5, each person appearing in the image is sequentially highlighted and queried by Qwen2.5-VL-72B (Bai et al., 2025b) with two prompts. The first prompt requests a detailed analysis of the individual's emotions and thoughts to produce the intermediate output. The intermediate output is provided to Qwen3 (Yang et al., 2025) to yield an identity-neutral emotional analysis. The second prompt seeks a comprehensive description of the person's behaviors, interactions, and any plausible intentions, resulting in a behavior description. This description is passed to Qwen3 (Yang et al., 2025) to produce the final intention analysis, past-cause description and future-consequence description. This helps prevent same-model bias by using an independent text-only model for final text generation.

## 2.3 Manual Quality Review and Adjustment

To refine the automatically generated annotations, we design a custom Gradio-based (Abid et al., 2019) interface supporting cluster-level de-duplication and instance-level correction. A detailed description of the interface and workflow is provided in Appendix C.

At the cluster de-duplication stage, experts select representative and diverse samples for each group of similar images that have been auto-annotated. They are expected to find the images with best automatic annotation quality, best image quality and highest complexity. This step helps us maximize image diversity while better utilizing annotation results on similar images. At the instance correction stage, experts adjust bounding boxes and attributes with real-time visualization. Experts can decide whether to accept or discard each image and make detailed modifications to the automatic annotation results, especially the features that rely on Qwen2.5-VL-72B to annotate. This step contributes

to higher annotation quality and eliminates potential same-model bias (Panickssery et al., 2024) of Qwen models which are among the models evaluated.

## 2.4 QUESTION-ANSWER DESIGN

We design 21 question types based on the annotated features, covering eight dimensions: Face Understanding (FU), Body Understanding (BU), HOI Understanding (HU), Multi-Image Understanding (MIU), Multi-Person Reasoning (MPR), Intention Discrimination (ID), Causal Discrimination (CD), and Emotion Discrimination (ED). The answer formats include Choice, Bounding Box, Short-Answer, Ranking, Judgment, as well as composite forms such as Judgment combined with Short-Answer, Judgment combined with Bounding Box, and Short-Answer combined with Bounding Box. Here we explain how to construct each of the five question components. Detailed construction logic and examples for eight dimensions and 21 question types are provided in Appendix D.

**Choice** questions, except for Causal Choice questions, have only one correct answer. The distinctive feature of Causal Choice questions is that they require selecting one past event (the cause) and one future outcome (the consequence) from four options. When constructing incorrect options, we prioritize confusing patterns. For example, in Wearing Choice, we select only the same type of wearing as incorrect options. In HOI Choice, we deliberately confuse left and right hands as body parts conducting interaction. In Emotion Analysis Choice, we choose emotion analyses from images where the subject's mood is broadly similar as incorrect options.

**Bounding Box** questions require MLLMs to provide bounding boxes coordinates for facial parts, body parts, or HOI objects within images. While dedicated detection models exist to address such problems, these tasks serve as the most direct measure of MLLMs' image-text grounding capability. Furthermore, many questions in this benchmark like Judgment Bounding Box and Identify Open HOI involve complex or indeterminate conditions that open-set object detector cannot handle.

**Short-Answer** questions are open-ended, asking for brief responses primarily regarding the names and colors of clothing items, the names of HOI objects, and general attributes such as gender, age group, emotion, and race. The ground truth for these nouns or adjectives is derived from automated labeling and manually refined by experts to minimize same-model bias (Panickssery et al., 2024).

**Ranking** questions provides three features and four images, and requires the model to rank the images according to the number of features present in the people shown. Multi-Face focuses on facial features, while Multi-Wearing focuses on clothing. This tests the model's ability to simultaneously cross-check multiple features across multiple images.

**Judgment** questions are combined with other types of questions and require MLLMs to proceed with an answer only if certain conditions are met; otherwise, it should refuse to answer. For example, a Judgment Bounding Box question asks the model to check whether there is a person in the image with a specific feature; if so, it must provide the bounding box of a certain body part of that person; if not, it should return [-1, -1, -1, -1] as the answer. Such questions are essentially about true/false judgments, but here they are designed to target hallucination of MLLMs on human-centric problems.

**Different evaluation metrics are used for each question component.** We evaluate the performance of the models using metrics tailored to each component of question (detailed definitions are provided in Appendix G). For Choice questions, accuracy is reported; for Short-Answer questions, a composite score combining BERT F1 (Zhang et al., 2020), embedding cosine similarity (Wang et al., 2020), and keyword coverage is used; for bounding-box questions, Intersection-over-Union (IoU) is used; for Ranking questions, Kendall's Tau (Kendall, 1938) is reported; and for judgment questions, F1 score balances precision and recall.

## 2.5 STATISTICS AND ANALYSIS

**Image Statistics.** Human-MME contains a diverse collection of images representing a wide range of real-world scenarios. Figure 3(a) shows a sunburst chart of image content, organized into four main domains: Daily Life, Work, Study, and Entertainment, with subcategories covering typical situations. For example, the Office subcategory under Work includes Meetings, Desk Work, and Reports. Figure 3(b) presents the distribution of image resolutions, with over 30% of images having 4K (3840×2160) resolution or higher. Figure 3(c) shows the distribution of the number of people

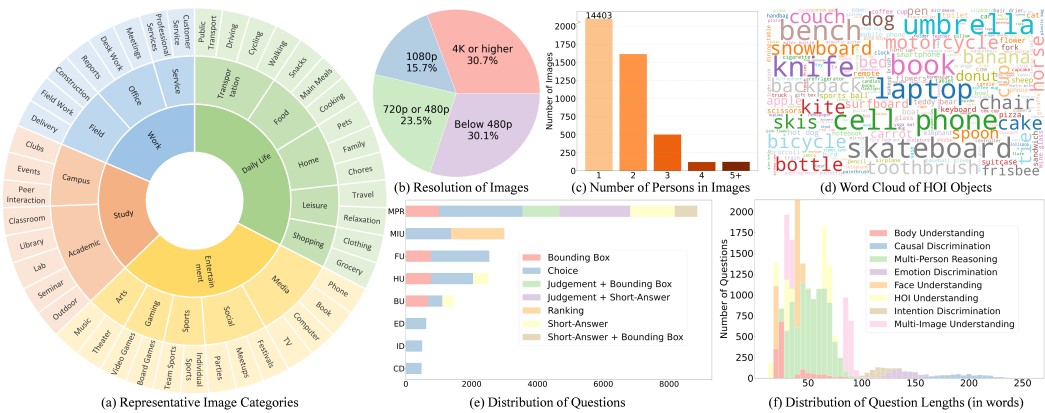

Figure 3: Human-MME demonstrates rich diversity: **Images:** (a) shows a sunburst chart of image content across four domains and subcategories; (b) indicates image resolutions ranging from below 480p to over 4K; (c) presents the number of people per image, from single- to multi-person scenes; (d) illustrates a word cloud of HOI objects, covering interactions with hundreds of distinct objects. **QA Pairs:** (e) illustrates multiple question types distributed across eight reasoning dimensions; (f) shows the lengths of questions, capturing variability in question complexity.

Table 1: Comparison to the existing benchmarks involving human features.

| Benchmark | Modality | #QA | Formats | Fine-grained grounding | Face Features | Body Features | Human-Object Interaction | Multi-image | Multi-person | High-level abstract features |
|---|---|---|---|---|---|---|---|---|---|---|
| MMBench | Image | 3.2K | Choice | ✗ | ✓ | ✗ | ✓ | ✗ | ✓ | ✓ |
| MME | Image | 2.8K | TF | ✗ | ✗ | ✓ | ✗ | ✗ | ✓ | ✗ |
| Seed-Bench | Image+Video | 19K | Choice | ✗ | ✗ | ✗ | ✓ | ✗ | ✓ | ✗ |
| HV-MMBench | Video | 8.7K | Choice/Open-ended/ TF | ✗ | ✗ | ✗ | ✗ | ✗ | ✓ | ✓ |
| HumanVBench | Video | 2.1K | Choice | ✗ | ✗ | ✓ | ✗ | ✗ | ✓ | ✓ |
| Face-Human-Bench | Image | 2.7K | Choice | ✗ | ✓ | ✗ | ✗ | ✓ | ✓ | ✓ |
| HumaniBench | Image | 32K | Choice/Open-ended/ BBox | ✗ | ✗ | ✓ | ✓ | ✗ | ✓ | ✓ |
| Human-MME (Ours) | Image | 20K | Choice/Open-ended/ TF/BBox/Ranking | ✓ | ✓ | ✓ | ✓ | ✓ | ✓ | ✓ |

per image, dominated by single-person scenes, with some multi-person scenes as well. Figure 3(d) illustrates a word cloud of HOIs, depicting interactions between humans and hundreds of different objects, such as skateboards, books, laptops and knives.

**QA Statistics.** Figure 3(e) demonstrates the distribution of question types across different dimensions. Among all dimensions, Multi-Person Reasoning contains the largest number of questions and the richest variety of answer formats. Choice questions are the most frequent answer format, appearing in all eight dimensions, with each question type containing at least 340 questions. Figure 3(f) presents the distribution of question lengths, reflecting variability in question complexity across dimensions. Questions in high-reasoning dimensions, such as Intention Discrimination, Causal Discrimination, and Emotional Discrimination, are generally longer, typically exceeding 100 words; in particular, Causal Discrimination questions have an average length of 200 words.

Table 1 compares Human-MME with existing human-related benchmarks. Human-MME provides larger scale and more diverse tasks, with richer formats and fine-grained annotations. It covers face and body features, HOI, multi-image and multi-person scenarios, and high-level abstract reasoning. Unlike Face-Human-Bench (Qin et al., 2025), which lacks fine-grained body details such as spatial grounding and has only one question format, and HumaniBench (Raza et al., 2025), whose bounding box tasks are coarse-grained and overlooks detailed human features, Human-MME is the first human-centric benchmark to offer a truly comprehensive and fine-grained evaluation. Further comparison in methodology of the benchmark construction can be found in Appendix A.

Table 2: **Human-MME scores by eight dimensions and five question components. Bold** indicates the best. underline indicates the second place. Closed-source models are ranked separately. Detailed results are provided in Appendix H.

| Model | FU | BU | HU | MIU | MPR | ID | CD | ED | Avg. | Bounding Box | Choice | Short-Answer | Ranking | Judgment |
|---|---|---|---|---|---|---|---|---|---|---|---|---|---|---|
| GLM-4.5V | **61.6** | **77.4** | **82.5** | 79.2 | **71.5** | 83.9 | 85.4 | 66.6 | **76.0** | **66.3** | 70.8 | **83.5** | 86.2 | 68.3 |
| GLM-4.1V-9B | 55.2 | 74.1 | 69.5 | 71.8 | 64.3 | 82.7 | 76.0 | 58.8 | 69.1 | 49.7 | 68.0 | 80.7 | 82.5 | 66.3 |
| Qwen2.5-VL-72B | 61.1 | 70.2 | 70.6 | 75.4 | 65.2 | **88.1** | **86.3** | 65.3 | 72.8 | 50.8 | 70.4 | 81.7 | 83.9 | **71.3** |
| Qwen2.5-VL-32B | 56.2 | 73.3 | 65.3 | 70.7 | 58.2 | 82.9 | 81.1 | 64.9 | 69.1 | 44.9 | 67.9 | 72.7 | 82.4 | 67.0 |
| Qwen2.5-VL-7B | 49.4 | 68.4 | 61.4 | 61.0 | 46.3 | 84.1 | 72.1 | 60.9 | 63.0 | 31.7 | 60.1 | 71.0 | 70.7 | 56.5 |
| Intern-S1 | 41.0 | 65.2 | 65.5 | **79.8** | 59.3 | 82.9 | 83.2 | **68.3** | 68.2 | 22.1 | **72.6** | 82.0 | **86.6** | 68.9 |
| InternVL3.5-241B | 50.7 | 74.6 | 71.4 | 76.4 | 59.4 | 83.7 | 82.5 | 66.4 | 70.6 | 46.3 | 68.9 | 81.3 | 84.0 | 57.3 |
| InternVL3-78B | 43.4 | 67.9 | 67.2 | 78.6 | 54.6 | 86.7 | 84.7 | 67.7 | 68.9 | 26.6 | 70.9 | 82.9 | 85.2 | 61.6 |
| InternVL3.5-38B | 44.6 | 72.6 | 64.6 | 75.0 | 53.8 | 86.9 | 78.0 | 65.6 | 67.6 | 30.6 | 67.9 | 80.7 | 82.6 | 62.0 |
| Llama-4-Scout | 27.3 | 50.6 | 49.4 | 48.9 | 33.9 | 66.5 | 57.1 | 50.4 | 48.0 | 6.4 | 47.9 | 69.5 | 71.0 | 38.6 |
| LLaVA-NeXT-72B | 38.0 | 66.8 | 65.1 | 54.8 | 47.2 | 77.0 | 70.5 | 54.6 | 59.3 | 29.8 | 58.2 | 72.8 | 61.1 | 52.3 |
| Aya-vision-32B | 30.9 | 57.2 | 57.1 | 67.9 | 42.8 | 76.2 | 71.8 | 57.4 | 57.7 | 8.9 | 57.7 | 78.5 | 75.7 | 53.8 |
| Gemma3-27B | 35.1 | 59.9 | 61.2 | 65.3 | 45.1 | 81.5 | 73.0 | 60.1 | 60.2 | 13.8 | 59.4 | 78.4 | 75.4 | 54.5 |
| Kimi-VL-A3B | 37.3 | 63.1 | 50.8 | 27.3 | 42.6 | 81.0 | 63.1 | 55.3 | 52.6 | 17.2 | 56.7 | 72.1 | 63.2 | 50.4 |
| MiniCPM-V-4.5 | 38.9 | 62.6 | 62.4 | 73.5 | 52.1 | 81.5 | 67.8 | 63.3 | 62.8 | 20.2 | 65.0 | 80.7 | 84.0 | 57.9 |
| Phi-4 | 29.5 | 48.1 | 48.6 | 39.6 | 29.6 | 62.9 | 38.1 | 46.4 | 42.9 | 5.5 | 45.5 | 62.4 | 54.4 | 19.8 |
| Gemini-2.5-Pro | **42.4** | 66.5 | 70.0 | **83.6** | 48.6 | 79.4 | 86.1 | 64.5 | 67.6 | 23.5 | 72.4 | **83.9** | **90.9** | **72.0** |
| GPT-4o | 28.8 | 58.8 | 59.8 | 74.7 | 34.4 | 79.2 | 76.2 | 52.7 | 58.1 | 11.5 | 57.6 | 78.3 | 83.8 | 48.6 |
| GPT-5 | 34.4 | **67.8** | **71.1** | 75.8 | **43.1** | 82.3 | **89.2** | 42.6 | 63.3 | **25.2** | 65.8 | 77.5 | 85.8 | 41.7 |
| GPT-5-mini | 30.6 | 66.3 | 67.4 | 76.4 | 41.3 | 79.4 | 81.7 | 39.9 | 60.4 | 21.1 | 63.4 | 76.9 | 87.4 | 49.1 |

# 3 EXPERIMENTS

## 3.1 EXPERIMENTAL SETTING

**Evaluated MLLMs.** We conduct evaluations on several vision-language models, including GLM-4.5V (Hong et al., 2025), GLM-4.1V-9B (Hong et al., 2025), Qwen2.5-VL (72B, 32B, 7B, (Bai et al., 2025b)), InternVL3-78B (Zhu et al., 2025), InternVL3.5 (38B, 241B) (Wang et al., 2025), Intern-S1 (Bai et al., 2025a), MiniCPM-V-4.5 (Yao et al., 2024), Gemma3-27B (Kamath et al., 2025), LLaVA-NeXT-72B (Chen & Xing, 2024), Aya-vision-32B (Dash et al., 2025), Kimi-VL-A3B (Du et al., 2025), Llama-4-Scout (Meta AI, 2025), and Phi-4 (Abouelenin et al., 2025). More details of these open-source models are provided in Appendix E. In addition, we also evaluated four closed-source models: Gemini-2.5-Pro (Comanici et al., 2025), GPT-4o (Hurst et al., 2024), GPT-5, and GPT-5-mini (OpenAI, 2025).

**Implementation Details.** We use VLLM (Kwon et al., 2023) to deploy the open-source MLLMs under evaluation, and for the closed-source MLLMs we call their official APIs. To ensure structured and consistent outputs from the evaluated MLLMs, we employ format-enforcing prompts for each question type (see Appendix F). Model outputs are parsed using regular expressions to extract the final answers. This will also minimize the bias introduced by the generation style of MLLMs.

## 3.2 RESULTS

Table 2 compares performance across the eight evaluation dimensions. Models trained with explicit grounding data consistently lead on perception-oriented tasks. In particular, GLM-4.5V achieves the highest scores on most vision-heavy dimensions such as Face Understanding, Body Understanding, HOI Understanding and Multi-Person Reasoning, reflecting the value of its precise visual element localization. Qwen2.5-VL-72B performs almost as well in these areas and surpasses all others on higher-level reasoning tasks such as Intention and Causal Discrimination. By contrast, models without explicit grounding training, such as Gemma3-27B, Aya-vision-32B and Llama-4-Scout, lag far behind in localization-heavy tasks. An interesting exception is Intern-S1, which, despite lacking fine-grained grounding ability, reaches the top score in multi-image understanding and emotion discrimination, indicating strong high-level understanding abilities. Among the two closed-source models tested, Gemini-2.5-Pro consistently outperforms GPT-4o. Moreover, Gemini-2.5-Pro shows

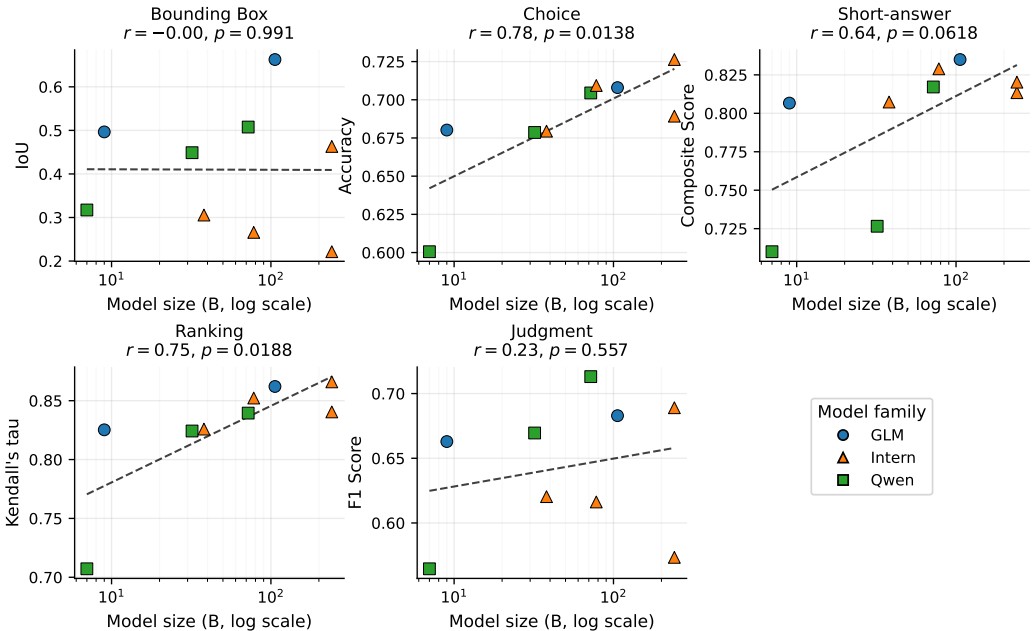

Figure 4: **Correlation between model size and performance in different question components.** To minimize the influence of differences in model architecture and training strategies, only the models ranking in the top half of the overall performance table (Table 2) are selected for this analysis. These models all belong to the GLM, Qwen, and Intern families.

a pattern very similar to Intern-S1: it lacks fine-grained image grounding ability but demonstrates strong high-level understanding abilities.

Table 2 also breaks down the results by five question components. GLM-4.5V leads in Bounding Box and Short-Answer related questions, confirming its advantage in spatial localization and consistent recognition. Meanwhile, Intern-S1 and InternVL3-78B excel at choice, short-answer and ranking questions but perform poorly on Bounding Box questions, highlighting their strengths in consistent human feature recognition and limits in fine-grained spatial alignment. Qwen2.5-VL-72B achieves the highest Judgment score, shows reliable selective answering, reflecting a strong ability to prevent hallucination. Among the closed-source models, Gemini-2.5-Pro excels in all aspects except for Bounding Box related questions, matching or surpassing the best open-source models.

### 3.3 FINDINGS OF HUMAN-MME

We have seven findings from this benchmark, which are listed here. More detailed analysis and discussion of these findings can be found in the Appendix I.

**Stronger scaling effects in Choice and Ranking tasks.** Figure 4 shows that performance on Choice and Ranking components have stronger correlation with model size than other metrics. Larger models can process and integrate multiple visual features simultaneously, which enhances their ability to evaluate candidate options and order items effectively. For details, see the discussion in Appendix I..

**Training data influence on grounding tasks.** Table 10 in Appendix E and Figure 4 show that Bounding Box performance is influenced more by the relevance and composition of training data than by model scale or model architecture. Models whose training sets explicitly include human-centric grounding examples, or that place emphasis on grounding-related supervision, achieve noticeably higher and more stable results. In addition, models that apply structured output-format alignment, such as using normalized or JSON-based bounding box representations with specialized tokens, further benefit from improved precision and consistency. These patterns indicate that grounding-focused data and clear output-format constraints are central factors in obtaining strong visual grounding performance. For details, see the discussion in Appendix I.

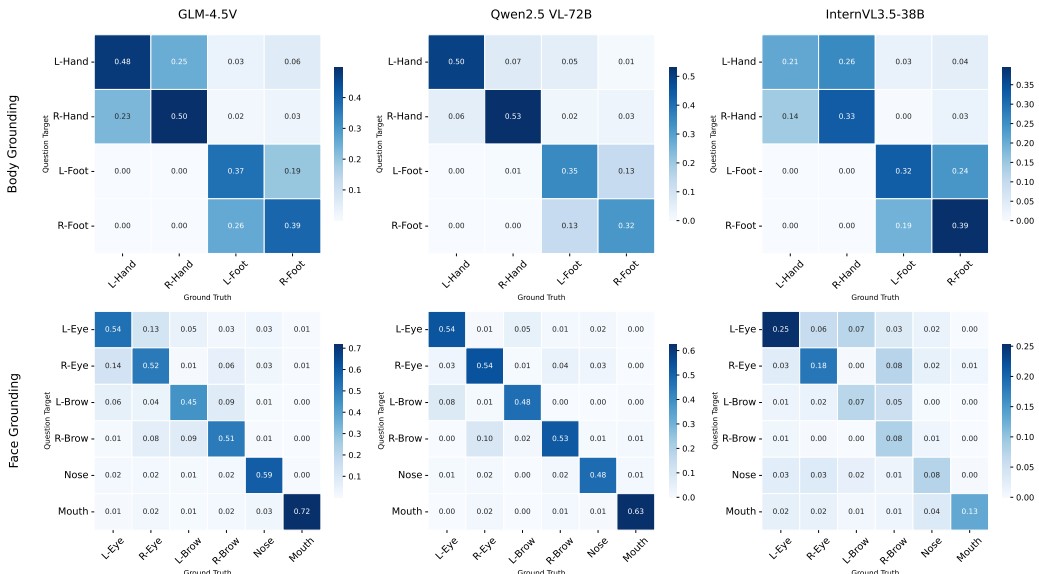

Figure 5: **Confusion matrices for body and face grounding tasks.** This figure presents the confusion matrices of three MLLMs on the Body Grounding and Face Grounding tasks. All images containing any overlap or ambiguity between left and right hands or feet are removed in advance to ensure that the evaluation focuses purely on the models' ability to distinguish left-right body and facial parts.

Table 3: P, R, F1 of different models in choosing whether to answer or abstain on questions with Judgment component and their performance compared to similar task without Judgment component.

| Model | Precision | Recall | F1 Score | Bounding Box | J + Bounding Box | Short-Answer | J + Short-Answer |
|---|---|---|---|---|---|---|---|
| GLM-4.5V | 54.7 | *90.9* | 68.3 | **65.8** | **63.8(-2.0)** | 81.6 | 80.1(-1.5) |
| GLM-4.1V-9B | 53.0 | 88.7 | 66.3 | 45.0 | *46.8(+1.7)* | 79.1 | 76.9(-2.2) |
| Qwen2.5-VL-72B | **60.2** | 87.6 | 71.3 | 57.3 | 49.7(-7.6) | *79.8* | 80.7(+0.9) |
| Qwen2.5-VL-32B | 53.5 | 89.4 | 67.0 | 46.4 | 43.2(-3.1) | 74.3 | 66.1(-8.1) |
| Qwen2.5-VL-7B | 48.8 | 68.0 | 56.5 | 33.2 | 23.5(-9.7) | 73.8 | 69.4(-4.4) |
| Intern-S1 | *57.3* | 86.5 | *68.9* | 22.6 | 19.9(-2.7) | 79.3 | 78.5(-0.8) |
| InternVL3-78B | 45.9 | *93.9* | 61.6 | 28.2 | 25.8(-2.4) | 79.4 | **81.9(+2.4)** |
| InternVL3.5-38B | 47.1 | 90.8 | 62.0 | 28.7 | 25.8(-3.0) | 77.9 | 78.2(+0.2) |
| Llama-4-Scout | 33.3 | 50.1 | 38.6 | 6.6 | 1.8(-4.8) | 67.9 | 62.4(-5.5) |
| LLaVA-NeXT-72B | 36.9 | 89.8 | 52.3 | 27.2 | 24.8(-2.4) | 68.8 | 71.1(+2.3) |
| Aya-vision-32B | 38.9 | 88.1 | 53.8 | 8.7 | 6.3(-2.4) | 75.8 | 74.4(-1.4) |
| Gemma3-27B | 37.8 | **98.1** | 54.5 | 15.5 | 15.3(-0.2) | 77.0 | 77.7(+0.7) |
| Kimi-VL-A3B | 41.4 | 67.0 | 50.4 | 19.5 | 19.6(+0.1) | 73.1 | 72.2(-0.9) |
| MiniCPM-V-4.5 | 42.8 | 89.9 | 57.9 | 24.5 | 20.5(-4.0) | 79.4 | 78.6(-0.8) |
| Phi-4 | 42.2 | 16.9 | 19.8 | 4.1 | 0.7(-3.4) | 66.9 | 59.9(-7.1) |
| GPT-4o | 40.5 | 61.5 | 48.6 | 10.8 | 6.9(-3.9) | 75.9 | 70.6(-5.3) |
| Gemini-2.5-Pro | **60.1** | 89.8 | 72.0 | 23.8 | 19.1(-4.7) | 82.2 | 80.4(-1.8) |

**Challenges in left-right discrimination for body parts.** Figure 5 shows that all evaluated models experience consistent difficulty in distinguishing left from right for hands and feet, often confusing the two sides across a range of poses and viewpoints. In contrast, their left-right discrimination on facial features, including eyes and eyebrows, is markedly more reliable. This discrepancy suggests that the spatial configuration of facial components provides more stable and unambiguous cues during training and inference, whereas the variable placement and articulation of limbs introduce greater ambiguity for MLLMs. For details, see the discussion in Appendix I.

**Precision-recall tradeoff in Judgment tasks.** Table 3 shows that models generally reach high recall and relatively low precision, often failing to abstain when no valid target exists. This indicates a tendency toward hallucination. Models with stronger mechanisms to prevent hallucination improve precision but reduces recall and sometimes leads to over-cautious refusals, reflecting a tradeoff between cautiousness and faithful instruction following. For details, see the discussion in Appendix I.

**Extra Judgment component reduces task performance.** Table 3 in Appendix I further shows that adding a Judgment component to questions lowers performance. The need to match two specified features before answering increases the complexity and reduces accuracy despite extra hint provided. For details, see the discussion in Appendix I.

**Hierarchy of discrimination difficulty.** Table 2 in Section 3.2 shows that accuracy generally follows the pattern Intention > Cause > Emotion, reflecting an increasing level of abstraction and difficulty across these discrimination tasks. For details, see the discussion in Appendix I.

Table 4: Total refusal rate (%).

| Model | FU | BU | HU | MIU | MPR | ID | CD | ED | Avg. |
|-------|----|----|----|-----|-----|-----|-----|-----|------|
| Gemini-2.5-Pro | 0.0 | 0.0 | 0.0 | 0.0 | 0.0 | 0.0 | 0.0 | 0.0 | 0.0 |
| GPT-4o | 0.0 | 0.0 | 0.0 | 0.0 | 0.0 | 0.0 | 0.0 | 0.0 | 0.0 |
| GPT-5 | 5.1 | 0.0 | 0.0 | 0.6 | 2.6 | 1.2 | 0.0 | 16.6 | 3.3 |
| GPT-5-mini | 10.2 | 0.0 | 0.0 | 0.6 | 6.1 | 4.0 | 0.2 | 25.5 | 5.8 |

Table 5: Refusal rate due to policy (%).

| Model | FU | BU | HU | MIU | MPR | ID | CD | ED | Avg. |
|-------|----|----|----|-----|-----|-----|-----|-----|------|
| Gemini-2.5-Pro | 0.0 | 0.0 | 0.0 | 0.0 | 0.0 | 0.0 | 0.0 | 0.0 | 0.0 |
| GPT-4o | 0.0 | 0.0 | 0.0 | 0.0 | 0.0 | 0.0 | 0.0 | 0.0 | 0.0 |
| GPT-5 | 2.1 | 0.0 | 0.0 | 0.3 | 1.7 | 0.0 | 0.0 | 7.3 | 1.4 |
| GPT-5-mini | 3.8 | 0.0 | 0.0 | 0.6 | 3.2 | 0.0 | 0.2 | 6.1 | 1.7 |

Table 6: Bounding box IoU comparison between the proprietary models, best open-source model (GLM-4.5V), and slightly weaker open-source model (InternVL3.5-38B) for face and body parts.

| Model | Body | | | | | Face | | | | | | |
|-------|------|------|------|------|------|------|------|------|------|------|------|------|
| | Left hand | Right hand | Left foot | Right foot | Whole body | Left eye | Right eye | Left brow | Right brow | Nose | Mouth | Whole face |
| GLM-4.5V | 0.497 | 0.513 | 0.394 | 0.400 | 0.964 | 0.540 | 0.517 | 0.447 | 0.513 | 0.594 | 0.719 | 0.887 |
| InternVL3.5-38B | 0.259 | 0.360 | 0.330 | 0.393 | 0.640 | 0.253 | 0.184 | 0.074 | 0.082 | 0.077 | 0.133 | 0.415 |
| Gemini-2.5-Pro | 0.129 | 0.144 | 0.066 | 0.056 | 0.688 | 0.066 | 0.058 | 0.049 | 0.034 | 0.160 | 0.121 | 0.474 |
| GPT-4o | 0.052 | 0.047 | 0.021 | 0.037 | 0.346 | 0.000 | 0.005 | 0.006 | 0.008 | 0.016 | 0.024 | 0.242 |
| GPT-5 | 0.176 | 0.195 | 0.090 | 0.100 | 0.739 | 0.023 | 0.028 | 0.014 | 0.051 | 0.062 | 0.085 | 0.547 |
| GPT-5-mini | 0.109 | 0.148 | 0.104 | 0.079 | 0.596 | 0.028 | 0.026 | 0.025 | 0.030 | 0.044 | 0.021 | 0.406 |

**Proprietary models underperform strong open-source systems.** As shown in Table 4 and Table 5, safety alignment affects GPT-5 and GPT-5-mini in particular, but this factor alone cannot account for the overall gap. The main weakness lies in bounding box tasks, where proprietary models handle whole-face and whole-body regions adequately but fail sharply on fine-grained facial and body-part localization, revealed by Table 6. This pattern suggests limited grounding capability at detailed spatial scales, possibly due to imbalanced training data. GPT-4o is further constrained by its older model, causing limited performance (Fu et al., 2024b; Lu et al., 2024; Zhang et al., 2024b). For details, see the discussion in Appendix I.

# 4 CONCLUSION

In this work, we presented **Human-MME**, a comprehensive benchmark specifically designed to evaluate the human-centric perception and reasoning abilities of multimodal large language models. Our benchmark integrates a rich spectrum of tasks, spanning from fine-grained facial and body understanding to high-level intention, causal, and emotional reasoning. By coupling an automated annotation pipeline with a rigorous manual review process, we ensure both scalability and the high fidelity of annotations, while supporting diverse question-answer formats such as multiple-choice, short-answer, grounding, ranking, and judgment-based tasks. The extensive experiments on 20 state-of-the-art MLLMs effectively expose the limitations and guide future MLLMs toward better human-centric image understanding and reasoning. We hope that the proposed benchmark, analyses, and insights will serve as a foundation and catalyst for the next generation of multimodal systems that more deeply and reliably comprehend human scenes and behaviors.

# 5 REPRODUCIBILITY STATEMENT

We have already elaborated on all the models or algorithms proposed, experimental configurations. and benchmarks used in the experiments in the main body or appendix of this paper. Furthermore. we declare that the entire code used in this work will be released after acceptance.

## 6  ETHICS STATEMENT

**Data Privacy and Licensing.** The Human-MME benchmark utilizes images from Pexels, Pixabay, and the HICO-DET dataset. We strictly adhere to the licensing terms of all data sources. Images from Pexels and Pixabay are used under their respective Content Licenses (Pexels, 2026; Pixabay, 2026), which permit free worldwide usage, modification, and distribution for academic and commercial purposes without mandatory attribution. In accordance with these licenses, we ensure that: (1) the images are not redistributed or sold as standalone copies without significant creative modification; (2) identifiable individuals are not portrayed in an offensive, immoral, or misleading manner. For HICO-DET, we comply with its original distribution policy (Chao et al., 2018). To further safeguard personal privacy, we will promptly remove specific images from the benchmark upon receipt of a valid request from the depicted individuals.

**Annotation Process.** The manual correction and quality assurance process was conducted entirely by the authors of this paper. Utilizing our expert knowledge of multimodal reasoning and human-centric computer vision, we performed a rigorous review of the automatically generated labels. This internal annotation strategy ensured a high level of consistency and professional judgment in handling complex scenarios, such as disambiguating fine-grained body parts and interpreting higher-dimensional causal relations. No external crowdsourcing platforms or underpaid labor were involved in the creation of this benchmark.

ACKNOWLEDGMENTS

This work was supported by the National Natural Science Foundation of China (No. 62320106007) and the NUS Start-up Grant (A-0010106-00-00).

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

APPENDIX

CONTENTS

## A  RELATED WORK

**Multimodal Large Language Models.** The emergence of large language models (LLMs) (Touvron et al., 2023; Achiam et al., 2023) has driven substantial progress in the development of multimodal large language models (MLLMs) (Bai et al., 2025b; Chen et al., 2024; Hurst et al., 2024; Li et al., 2023b; Liu et al., 2024a; Georgiev et al., 2024; Ye et al., 2024) for visual-language understanding. The progress is boosted from the BERT-based language decoders and progressively integrating developments in LLMs (Yin et al., 2024). Multimodal large language models (MLLMs) demonstrate improved capabilities and performance, largely due to end-to-end training techniques that leverage advanced LLMs such as GPT series (Achiam et al., 2023; Brown et al., 2020), LLaMA (Touvron et al., 2023), Alpaca (Taori et al., 2023), PaLM (Chowdhery et al., 2023; Anil et al., 2023), BLOOM (Muennighoff et al., 2022), and Vicuna (Chiang et al., 2023). Recent model advances—including Flamingo (Awadalla et al., 2023), PaLI (Laurençon et al., 2023), PaLM-E (Driess et al., 2023), BLIP-2 (Li et al., 2023b), Phi-4 (Abouelenin et al., 2025), Otter (Li et al., 2025) , MiniGPT-4 (Zhu et al., 2023) , MiniCPM-V (Yao et al., 2024), LLaVA-NeXT (Chen & Xing, 2024), Qwen-VL (Bai et al., 2025b), GLM-4V (Hong et al., 2025), and InternVL (Wang et al., 2025) introduce novel approaches to obstacles such as improving instruction-following abilities, addressing alignment problems, and scaling up the pretraining model. Nevertheless, the performance of these models in fine-grained human-centric image perception often remains underexplored.

**Multimodal Large Language Model Benchmarks.** In the field of MLLMs, numerous benchmarks have been developed to evaluate models' capabilities in both perception and cognition. As these MLLMs (Liu et al., 2024b; Yu et al., 2023) have demonstrated exceptional performance in general perception tasks, benchmarks regarding scientific understanding (Li et al., 2024), multimodal mathematical reasoning (Lu et al., 2024; Zhang et al., 2024a), and multi-disciplinary (Yue et al., 2024; Fu et al., 2024a; Li et al., 2023a; Chen et al., 2025; Jiang et al., 2026; Gong et al., 2025) capabilities have drawn increasing attention. Among these benchmark, there exist some attempts to analyze the temporal and dynamic of human-centric video understanding (Cai et al., 2025; Zhou et al., 2024), which assign great attention to temporal perceptions and ignore fine-grained human-centric perceptions. There are also benchmarks focusing on human-centric image understanding (Qin et al., 2025; Raza et al., 2025). However, (Qin et al., 2025) has a question format that is monotonous and lacks fine-grained features, and its scale is limited, while overly focusing on facial aspects. Raza et al. (2025) lacks questions related to facial features. And both of them do not have fine-grained grounding features. Figure 1 shows comparison between our work and previous human-related benchmarks. To address this gap, we propose a dedicated benchmark focusing on human-centric image comprehension, which systematically evaluates model performance from granular human comprehension to high-order intricate spatial and causal reasoning perception.

Table 7: Methodology comparison to the existing benchmarks involving human features.

| Benchmark | Scale | Data source | Labeling | QA curation | Quality assurance |
|---|---|---|---|---|---|
| MMBench | 3.2K | public datasets and Internet | No labeling, QA are constructed directly | Human | Human verification after multiple LLMs filter |
| MME | 2.8K | public datasets | No labeling, QA are constructed directly | Human | Not mentioned |
| Seed-Bench | 19K | public datasets | captions of datasets and text generated by foundation models | GPT-4 generate | Human verification |
| HV-MMBench | 8.7K | Internet | Qwen2.5 | Qwen2.5 generate | Human verification after Qwen2.5 filter |
| HumanVBench | 2.1K | Internet | Video-MLLMs generate | LLMs generate | Human verification |
| Face-Human-Bench | 2.7K | public datasets | captions of datasets | templates | Human verification |
| HumaniBench | 32K | Internet | GPT-4o | templates | Human verification |
| Human-MME (Ours) | **20K** | public datasets and Internet | **various specialized models** and LLMs | templates | Human **filtering and correction** |

As shown in Table 7, our benchmark design offers several advantages. First, by focusing specifically on the domain of human images, we can rely on a broad set of reliable expert models tailored for human-centric analysis rather than depending heavily on manual annotation, preexisting dataset labels, or LLM-generated labels. Manual annotation is constrained by cost, existing dataset labels risk information leakage and limit flexibility in question construction, and LLM annotation may introduce bias for certain question types and cannot reliably assess abilities such as fine-grained

grounding. Second, by extracting rich portrait features using multiple specialized models, we obtain highly structured and comprehensive representations of each individual. This enables integrated human-centric reasoning that is not restricted to dataset-provided labels or to the limited set of attributes observed by a single MLLM. Consequently, we can construct complex multi-step questions that combine multiple attributes from the same image, for example "What is the age group of the person who is wearing a beige jumpsuit". Because the data are structured, we do not rely on large language models for QA construction; instead, templates allow efficient generation of large volumes of category-specific QA pairs. Third, we exploit the fact that the data source contains many visually similar images. By aggregating and comparing labels across similar instances, we selectively retain high-quality annotations, which greatly improves the efficiency of human verification while ensuring diversity across the dataset.

# B  AUTOMATED ANNOTATION DETAIL

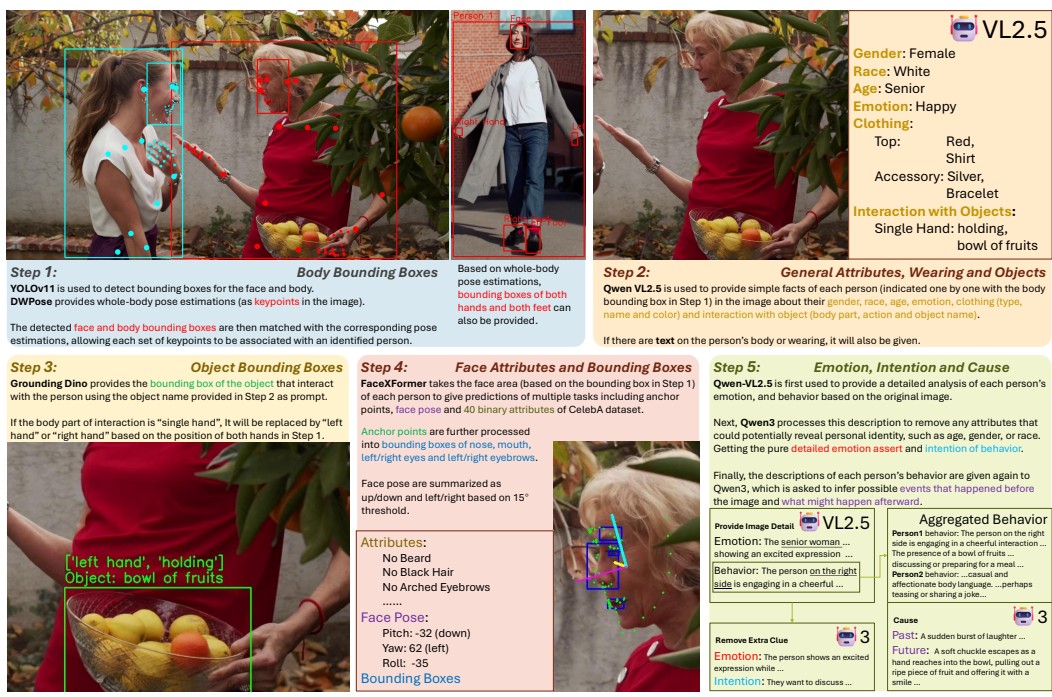

Figure 6: Annotation Pipeline before Manual Correction and QA Generation, corresponding to Section 2.2.

Table 8: Overview of extracted feature sets and their meanings.

| Symbol | Description |
|---|---|
| $B_i$ | Bounding box of the full body of person $i$ |
| $F_i$ | Bounding box of the face of person $i$ |
| $P_i^{\text{name}}$ | Bounding box of specific body part (*e.g.*, left hand, right foot) of person $i$ |
| $A_i^{\text{name}}$ | General attribute (*e.g.*, gender, age group, race) of person $i$ |
| $W_i^{(j)}$ | Wearing item of person $i$, including type, color, and name |
| $O_i^{(j)}$ | Object interaction of person $i$, including object and bounding box, action and body part |
| $FA_i^{\text{name}}$ | Facial attribute of person $i$ (*e.g.*, expression, hair style, presence of beard) |
| $FP_i^{\text{name}}$ | Region-specific facial part box of person $i$ (*e.g.*, nose, mouth, left eye) |
| $E_i$ | Identity-neutral emotional analysis of person $i$ |
| $I_i$ | Identity-neutral intention inferred from the behavior of person $i$ |
| $C^{\text{past}}, C^{\text{future}}$ | Scene-level cause (past) and consequence (future) narratives |

Figure 6 shows details about the automatic annotation pipeline applied to the collected dataset. The automatic annotation result will be used to generate question-answer pairs after manual selection and correction.

In Step 1 of Figure 6, we begin by applying DWPose (Yang et al., 2023) to each image to obtain whole-body pose estimates per person instance with 134 keypoints, including 18 body keypoints, 21 keypoints for each hand, 68 facial keypoints, and 3 keypoints for each foot (each with a confidence score). In parallel, a pre-trained YOLOv11 detector produces candidate body and face bounding boxes.

For each DWPose instance, we align the body box by selecting the detection that has the greatest overlap with the minimum bounding rectangle (MBR) of the keypoints. Formally:

$$B_i = \arg\max_{B \in \mathcal{B}} \text{IoU}\big(B,\ \text{MBR}(K_i)\big)$$

Here, $\mathcal{B}$ is the set of YOLOv11 (Jocher & Qiu, 2024) body boxes; $K_i$ is the keypoint set of the $i$-th DWPose (Yang et al., 2023) instance; $B_i$ denotes the matched body box for person $i$. Face matching is analogous (replacing $\mathcal{B}$ by the face set $\mathcal{F}$) and yields a matched face box $F_i$. When a reliable body-pose match cannot be established, the image is discarded at this stage; when a face match is unavailable, we simply retain body-only or face-only cases to accommodate close-ups and occlusions.

For body parts, if the minimum confidence over hand/foot keypoints is high, we derive part-level boxes from their keypoints' MBRs. We denote any such part box for person i by $P_i^{name}$, where name is one of "left hand", "right hand", "left foot", or "right foot".

In Step 2 of Figure 6, we use the matched body bounding box $B_i$ obtained from Step 1 to isolate each person instance within the image. For each instance, we query the vision-language model Qwen VL2.5-72B Bai et al. (2025b) to extract a set of factual attributes. The prompts used for this querying process follow a templated format, detailed in the appendix.

The model outputs are parsed to obtain three categories of information:

- General Attributes: These include perceived properties such as gender, age group, and race. We denote the general attributes of person $i$ as $A_i^{gender}$, $A_i^{age}$, $A_i^{emotion}$ and $A_i^{race}$.
- Wearing Attributes: These describe the clothing and accessories worn by the person, including their type (*e.g.*, "top"), color (*e.g.*, "red"), and name (*e.g.*, "shirt"). Each identified item is represented as $W_i^{(j)}$, indicating the $j$-th wearing item for person $i$.
- Object Interaction: The model also predicts interactions between the person and nearby objects. Each interaction is expressed as a triplet $O_i^{(j)}$, where the object name (*e.g.*, "bowl of fruits"), the interacting body part (*e.g.*, "single hand"), and the type of interaction (textite.g., "holding") are all inferred.

In Step 3 of Figure 6, the original image and the corresponding object name $O_i^{(j)}$ are input to Grounding DINO (Liu et al., 2023), which predicts the bounding box for each object. If the body part assigned to $O_i^{(j)}$ in Step 2 is "single hand", it is further refined at this stage. Since DWPose provides keypoints for both the left and right hands in $K_i$, we compute the average distance from each hand's keypoints to the object bounding box. The hand with the smaller average distance is then used to replace "single hand" with either "left hand" or "right hand".

In Step 4 of Figure 6, for each individual, we first enlarge the face bounding box $F_i$ by a factor of two and crop the corresponding face region from the image. The cropped region is then passed to FaceXFormer (Narayan et al., 2024) for facial feature recognition, landmark localization, and head pose estimation. FaceXFormer predicts 40 binary facial attributes, all derived from the CelebA dataset (Liu et al., 2015). We define two probability thresholds: a high threshold and a low threshold. If the predicted probability for an attribute exceeds the high threshold, the face is considered to possess the corresponding attribute; if it falls below the low threshold, the attribute is considered absent; otherwise, the attribute is marked as uncertain. The extracted facial attributes are denoted as $FA_i^{name}$, where name refers to the specific attribute (*e.g.*, "goatee").

Among the head pose parameters provided by FaceXFormer, we focus on pitch and yaw, which define the face orientation in degrees. Using a threshold of $\pm15°$, we discretize pitch into up or down and yaw into left or right, where the latter indicates the side of the image toward which the face is oriented (rather than the subject's own left or right). These head pose attributes are also incorporated into $FA_i^{\text{up}}$, $FA_i^{\text{down}}$, $FA_i^{\text{left}}$, and $FA_i^{\text{right}}$ as part of the facial attributes.

In addition, FaceXFormer outputs 68 facial landmarks. Based on prior knowledge of these landmarks, we can derive bounding boxes for specific facial regions—including the nose, mouth, left eye, right eye, left eyebrow, and right eyebrow. The bounding box from FaceXFormer outputs will also be compared with the the facial part of DWPose results $K_i$ in Step 1 to see if they aligned well enough. When FaceXFormer results have low IoU compared to DWPose result, the bounding boxes will not be recorded. These region-specific bounding boxes are denoted as $FP_i^{\text{name}}$, where name refers to the corresponding facial part.

In Step 5 of Figure 6, the objective is to extract higher-level semantic attributes such as Emotion, Intention, and Cause. Each person appearing in the image is sequentially highlighted and queried by Qwen2.5-VL-72B with two prompts. The first prompt requests a detailed analysis of the individual's emotions and thoughts to produce the intermediate output $E_i^+$ using $A_i^{\text{emotion}}$ as reference. The second prompt seeks a comprehensive description of the person's behaviors, interactions with other people and objects, and any plausible intentions, resulting in a behavior description denoted as $I_i^+$.

Subsequently, $E_i^+$ is provided to Qwen3, which removes any information that could reveal personal identity, such as gender or age, yielding a purely identity-neutral emotional analysis $E_i$. In parallel, $I_i^+$ is passed to Qwen3 to perform two operations: first, to remove all identity-related details, and second, to explicitly extract any inferred intentions of the person. This process produces the final intention analysis $I_i$.

Finally, all behavior descriptions $I_i^+$ from every person in the image are aggregated and submitted to Qwen3 to infer the broader context of the scene. Qwen3 generates two high-level narratives: $C^{+\text{past}}$, describing possible events that may have occurred prior to the captured moment, and $C^{+\text{future}}$, predicting events likely to unfold afterward. Each of these narrative outputs is further processed by Qwen3 to remove identity-specific information, resulting in the final past-cause description $C^{\text{past}}$ and future-consequence description $C^{\text{future}}$.

The complete set of fine-grained features extracted for person $i$ can be expressed as:

$$\mathcal{T}_i = \left\{ B_i,\ F_i,\ \{P_i^{\text{name}}\}_{\text{name}},\quad \{A_i^{\text{name}}\}_{\text{name}},\ \{W_i^{(j)}\}_j,\ \{O_i^{(j)}\}_j, \{FA_i^{\text{name}}\}_{\text{name}},\ \{FP_i^{\text{name}}\}_{\text{name}} \quad \right\}$$

## C  MANUAL ANNOTATION USER INTERFACE

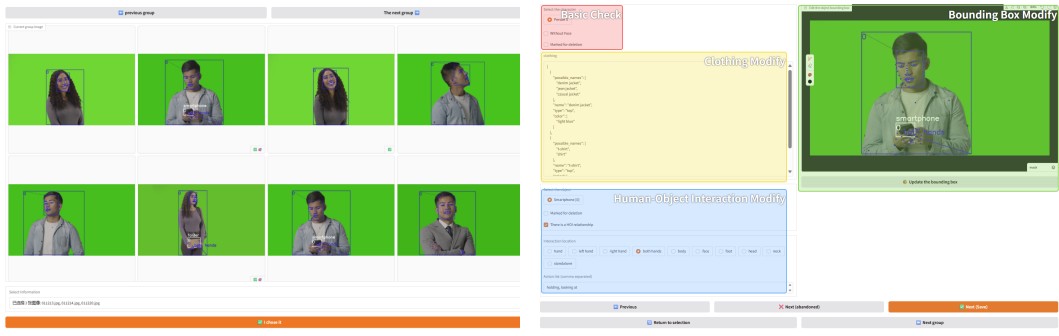

(a) Manual Selection and Deduplication.      (b) Manual Correction.

Figure 7: Manual Review Software Interface.

We developed a custom software tool based on Gradio (Abid et al., 2019) to facilitate both data deduplication and annotation refinement after automatic annotaation.

Because the de-duplication in Section 2.1 was relatively permissive, the automatic annotation process described in Section 2.2 inevitably produces redundant annotations for many nearly identical images. To address this, we first extract feature vectors with YOLOv11 and cluster the images accordingly. The subsequent annotation workflow then first operates at the cluster level.

As illustrated in Figure 7a, the first stage is Selection and De-duplication. Within each cluster of visually similar images, annotation experts are asked to select a subset that maximizes diversity of subjects and complexity of content. For example, when multiple images contain the same individuals, preference is given to those in which a HOI has been successfully annotated. Likewise, previewed bounding boxes should be as accurate as possible, ensuring that the automatic annotations capture all important individuals. After selecting the subset, the expert proceeds to the next step by clicking the button at the bottom of the interface, where each image is reviewed and corrected individually.

The Manual Correction phase involves a detailed inspection and refinement of the automatically generated annotations for every detected person. This includes removing or adjusting entire body bounding boxes when they are incorrectly assigned. These operations are performed in the red area of Figure 7b. Wearing attributes $W_i^{(j)}$ are examined and edited in the yellow area by directly modifying the corresponding JSON text. HOI attributes $O_i^{(j)}$ such as associated objects, body parts, or actions can be corrected in the blue area. Finally, bounding boxes can be adjusted in the green region within the red area, and all modifications are immediately visualized in real time in the same green region.

Once the corrections yield satisfactory results, the expert saves the manually refined annotations. If the automatic annotations are too poor to be corrected effectively, the expert can either return to the previous step to reselect images or discard the problematic image altogether. After completing all images in the current cluster, the process repeats for the next cluster until the entire dataset has been reviewed.

## D    QUESTION-ANSWER DETAIL

The Question-Answer Design stage leverages the rich set of features extracted during data annotation to construct a comprehensive evaluation framework. We design a total of 21 question types spanning eight dimensions: Face Understanding, Body Understanding, HOI Understanding, Multi-Image Understanding, Multi-Person Reasoning, Intention Discrimination, Causal Discrimination, and Emotion Discrimination. The corresponding answer formats cover seven distinct forms: single choice, ranking, short-answer, bounding box, judgment combined with short-answer, judgment combined with bounding box, and short-answer combined with bounding box. Representative examples of each question type are provided.

**Face Understanding**    applies to images containing only a single person $i$, we define two categories of questions: **Face Grounding** and **Face Choice**. In Face Grounding, the model is required to predict the bounding box of a specified facial region. The target region is randomly selected from the full face box $F_i$ or one of the finer-grained subregions $FP_i^{\text{mouth}}$, $FP_i^{\text{nose}}$, $FP_i^{\text{left eye}}$, $FP_i^{\text{right eye}}$, $FP_i^{\text{left eyebrow}}$, or $FP_i^{\text{right eyebrow}}$. In Face Choice, a single facial attribute is chosen from $FA_i$ as the correct answer. Three distractor attributes, absent from $FA_i$, are also provided, and the model must identify the correct one.

**Body Understanding**    also focuses on images with exactly one person $i$ and includes two question types: **Body Grounding**, **Wearing Choice** and **Wearing Short-Answer**. Body Grounding asks the model to predict the bounding box of a body region randomly selected from the full body box $B_i$ or one of the keypoint-based limb regions $P_i^{\text{left hand}}$, $P_i^{\text{right hand}}$, $P_i^{\text{left foot}}$, or $P_i^{\text{right foot}}$. Wearing Choice evaluates the model's understanding of clothing attributes: one wearing item is randomly sampled from $W_i$ as the correct answer, while three incorrect options are generated by altering its color and name. The question specifies the item's type, and the model must identify the correct wearing from the image. The replacement names and colors for the distractors are drawn from annotations of wearing items of the same type in other images. Wearing Short-Answer provides the type of a existing $W_i$ and ask model to provide its color and name.

**HOI Understanding** has three types of questions: **HOI Choice**, **HOI Short-Answer**, and **HOI Grounding**. For any person $i$ in the image, one of their object interactions $O_i$ is selected as the basis of the question. In HOI Choice, each option contains an object name, an action, and a body part; only one option is entirely correct, while the object name or body part in the remaining options is randomly altered. The model is required to select the correct option. In HOI Short-Answer, the action and body part are provided and the model must answer with the corresponding object name. In HOI Grounding, only the action and body part are given, and the model is expected to output the bounding box of the relevant object.

**Multi-Image Understanding** consists of three question types: **Multi-Face**, **Multi-Wearing**, and **Multi-HOI**. Multi-Face presents four images together with three facial attributes that may appear in $FA$. Among the four images, one contains a face that satisfies all three attributes, one satisfies two, one satisfies only one, and one satisfies none. The model is asked to rank the four images according to how many of the three attributes are satisfied by at least one face. Multi-Wearing follows the same logic but uses three clothing items that may appear in $W$; the model must order the images by the number of those clothing items present. Multi-HOI differs in that it provides a description including a body part, an action, and an object name; in the four candidate images some may have mismatched objects or body parts, and the model must identify the image that best matches the description.

**Multi-Person Reasoning** covers a broader set of question types and focuses on images containing multiple people. Questions typically target a specific person $i$ with feature set $\mathcal{T}_i$, while the other people in the image are indexed by $j$ and have feature sets $\mathcal{T}_j$.

- Identify-related questions: **Identify Short-Answer**, **Identify Bounding Box**, **Identify Open HOI**, and **Identify Choice**. They are constructed by first selecting a feature from $\mathcal{T}_i - \mathcal{T}_j$ that is unique to person $i$, ensuring that the model can localize the correct individual. For Identify Short-Answer, a second feature is chosen from $\mathcal{T}_i$ (such as clothing, HOI, or general attributes) and the model must provide the answer to a direct question about it. Identify Bounding Box instead selects a facial or body bounding box from $\mathcal{T}_i$ and requires the model to output the corresponding box. Identify Open HOI chooses an HOI from $O_i$ and provides its action and body part; the model must return both the object name and its bounding box. Identify Choice selects one feature from $\mathcal{T}_i$ as the correct option and three features from $\mathcal{T}_j - \mathcal{T}_i$ as distractors; the model must select the feature that matches person $i$.

- Judgement-based questions: **Judgement Short-Answer** and **Judgement Bounding Box**. They require the model to refrain from answering if no individual satisfies the specified criteria, and to proceed only when a suitable person exists. A unique feature $a$ is first drawn from $\mathcal{T}_i - \mathcal{T}_j$, and another feature $b$ is drawn from $\mathcal{T}_i \cup \mathcal{T}_j$, which may or may not belong to person $i$. If $b \in \mathcal{T}_i$, the question has a valid answer; if $b \notin \mathcal{T}_i$, the model is expected to explicitly decline to answer. The model is instructed to locate and focus only on a person who satisfies both features $a$ and $b$, and then provide the requested response, or state that no such person exists. Judgement Short-Answer subsequently asks about a feature from $\mathcal{T}_i$ (such as clothing, HOI, or general attributes). Judgement Bounding Box asks the model to output a facial or body bounding box from $\mathcal{T}_i$. These types of questions are specifically designed to detect model hallucinations.

- Finally, **Common Choice** is a single-choice question in which the correct answer is a feature drawn from $\bigcap_{i=1}^{n} \mathcal{T}_i$, *i.e.*, a feature shared by all people in the image. The three distractor options are features unique to individual persons. The model must select the feature that is common to every person present.

**Intention Discrimination** contains a single question type, **Intention Choice**. In this setting, an image features one person $i$ whose identity-neutral intention is denoted by $I_i$. Using CLIP, three visually similar images are retrieved, and the identity-neutral intentions from these three images are used as distractor options. The original image is presented to the model, which must select the intention description that best matches the depicted scene.

**Causal Discrimination** also includes only one question type, **Causal Choice**, which is a dual-selection task. For each image, there are two scene-level narratives: the cause $C^{\text{past}}$ and the con-

sequence $C^{\text{future}}$. CLIP is employed to find a visually similar image, from which $C'^{\text{past}}$ and $C'^{\text{future}}$ serve as incorrect alternatives. The model is required to determine which option correctly describes the past cause of the scene and which option correctly predicts its future consequence.

**Emotion Discrimination** comprises a single question type, **Emotion Analysis Choice**. Given an image with one person $i$ and that person's identity-neutral emotional analysis $E_i$, CLIP is used to locate visually similar images in which at least one person shares the same raw emotion label $A_i^{\text{emotion}}$. The identity-neutral emotional analyses of those individuals with matching raw emotions are taken as distractors. The model is then asked to choose the emotional analysis that most accurately reflects the emotions of the target person in the original image.

Representative examples for all the questions types are shown in Table 9.

Table 9: Examples of all question types.

| Question Type | Example Image | Example QA |
|---|---|---|
| **Face Grounding** |  | **Question:** Resolution of the image provided is 4096x2160. Please provide the bounding box of the facial part "mouth" of the main person in the image. 
 **Answer:** [2773, 1582, 3558, 1923] |
| **Face Choice** |  | **Question:** Please select the facial features of the person in the image from the following options (only one selection is allowed): 
 A. Wavy hair 
 B. Rosy cheeks 
 C. Has goatee 
 D. Mouth slightly open 
 **Answer:** D |
| **Body Grounding** |  | **Question:** Resolution of the image provided is 1920x1080. Please provide the bounding box (in xyxy format) of the "right hand" of the main person in the image. 
 **Answer:** [522,326,604,474] |
| **Wearing Choice** |  | **Question:** Please select the wearing of the person in the image from the following options (only one selection is allowed): 
 A. Black dress 
 B. Olive witch hat 
 C. Black bonnet 
 D. Black graduation gown 
 **Answer:** A |
| **Wearing Short-Answer** |  | **Question:** Please give name and color of the clothing item of type "headwear" that the main person is wearing in the image. 
 **Answer:** Party hat in pink and white |

*Continued on next page*

| Question Type | Example Image | Example QA |
|---|---|---|
| **HOI Choice** |  | **Question:** Please select the option that best describes the interaction between the person and object (only one selection is allowed): 
 A. body part: left hand action: holding object: beads 
 B. body part: right hand action: holding object: beads 
 C. body part: right hand action: holding object: flower branch 
 D. body part: left hand action: holding object: flower branch 
 **Answer:** D |
| **HOI Short-Answer** |  | **Question:** Please name the object that have interation "body part: left hand, action: holding" with the main person in the image. 
 **Answer:** cardboard box |
| **HOI Grounding** |  | **Question:** Resolution of the image provided is 4096x2160. Please provide the bounding box (in xyxy format) of the object that have interation "body part: left hand, action: holding" with the main person in the image. 
 **Answer:** [1619,701,2404,1472] |
| **Multi-Face** |  | **Question:** 
 - Narrow eyes 
 - Wearing earrings 
 - No beard 
 Listed are some facial attributes that appeared in the four images above. Please give the sequence of four images by the maximum count of facial attributes that appears in one single person. If someone in a specific image is showing all three facial attributes, it should be the first image in your answer, and if none of three facial attributes are present, it should be the last image. Please provide a explicit sequence of four images by their letters. 
 **Answer:** B - A - D - C |

*Continued on next page*

| Question Type | Example Image | Example QA |
|---|---|---|
| **Multi-Wearing** |  | **Question:**
- Gray trousers
- Gold necklace
- White sneakers
Listed are some clothing items that appeared in the four images above. Please give the sequence of four images by the maximum count of clothing listed that appears in one single person. If someone in a specific image is wearing all three clothing items, it should be the first image in your answer, and if none of three clothing items are present, it should be the last image. Please provide a explicit sequence of four images by their letters.
**Answer:** B - A - D - C |
| **Multi-HOI** |  | **Question:** This is a description of a human-object interaction: "body part: body, action: lie on, object: surfboard"
Which one of four images listed above best represents this interaction? Provide your answer with the corresponding image letter.
**Answer:** A |
| **Identify Short-Answer** |  | **Question:** There is one person in the image that meets the following condition:
- race: black
What is name and color of the clothing item of type "headwear" that the person is wearing?
**Answer:** Crown hat in red and green |

*Continued on next page*

| Question Type | Example Image | Example QA |
|---|---|---|
| **Identify Bounding Box** |  | **Question:** Resolution of the image provided is 3840x2160. There is one person in the image that meets the following condition:
- Have interaction with object:"body part: body, action: sitting on, object: chair"
Please provide the bounding box of the person's face in xyxy format.
**Answer:** [1954,1030,2208,1364] |
| **Identify Open HOI** |  | **Question:** Resolution of the image provided is 1920x1080. There is one person in the image that meets the following condition:
- Wearing Gray sweater
Please provide the name and bounding box in xyxy format of the object that have interation "body part: right hand, action: holding" with the person.
**Answer:** hiking poles [1005,694,1175,1076] |
| **Identify Choice** |  | **Question:** Resolution of the image provided is 4096x2160. There is one person in the image that meets the following condition:
- Wearing Sari in blue.
Ignoring other persons, please select the option that best describes the referred person.
A. Face turned to left side of image
B. Face turned to right side of image
C. face in the bounding box [2377,842,3009,1452] (xyxy format)
D. Wearing White necklace
**Answer:** B |
| **Judgement Short-Answer** |  | **Question:** There might be one person in the image that meets the following two conditions:
- Has beard
- Looking downward
Please answer the following question if there is such a person. Or else, please provide "unknown" as answer:
What is the gender of the person?
**Answer:** male |
| **Judgement Bounding Box** |  | **Question:** Resolution of the image provided is 1920x1080. There might be one person in the image that meets the following two conditions:
- Has goatee
- Face turned to left side of image
Please provide the bounding box of the person's face in xyxy format if there is such a person. Or else, please provide [-1,-1,-1,-1] as answer.
**Answer:** [1264,65,1446,367] |
| **Common Choice** |  | **Question:** Please select the option that fits most or all of persons in the image:
A. Wearing T shirt in white and black
B. gender: male
C. race: black
D. Not blond hair
Please provide the option letter of the most possible answer.
**Answer:** D |

| Question Type | Example Image | Example QA |
|---|---|---|
| **Intention Choice** |  | **Question:** Please select the best analysis of intention for someone appearing in the image:
A. The individual is expressing enthusiasm and engagement in a celebratory or performance-related activity through deliberate posing and vibrant attire
B. The individual is preparing for an outdoor adventure with companions and pets by organizing gear and coordinating plans while positioned near a vintage van on a mountain road
C. The individual is expressing joy and relaxation while engaging with a natural outdoor setting in a comfortable and effortless manner
D. The individual is expressing joy and contentment while embracing a relaxed and natural setting with confident ease
**Answer:** A |
| **Causal Choice** |  | **Question:** Please select the best analysis of what happened in the past and what will happen in the future:
A. The individual sets down the instrument, takes a deliberate breath, then flips through the pages with focused intent, adjusting finger placement and refining the rhythm before resuming with renewed precision.
B. The figure moved steadily along a winding trail, the earth beneath foot soft with fallen leaves and moss, drawn by the hush of the water and the distant whisper of reeds, pausing only to steady breath and release the weight of the day before stepping into the clearing.
C. The individual had carefully arranged the performance setup, positioning the instrument precisely on the surface, aligning the written material for optimal visibility, and connecting the digital device to the audio output, all with deliberate intent to begin a structured and focused session.
D. The hand moves with deliberate calm, tracing lines that breathe life into the stillness, each stroke a quiet declaration of presence, as the mind, unburdened and attuned, translates the essence of the moment into something tangible and enduring.
Please provide the option letters of the most possible answer separately for past and future.
**Answer:** past: C future: A |

| Question Type | Example Image | Example QA |
|---|---|---|
| **Emotion Analysis Choice** | | **Question:** Please select the best analysis of emotion for someone appearing in the image: 
 A. A deep sense of joy and contentment radiates from within as the individual experiences pure happiness and emotional ease in the moment 
 B. A deep sense of joy and contentment is present accompanied by a feeling of ease and fulfillment in the moment 
 C. A deep sense of accomplishment and fulfillment washes over with the realization of hard-earned success bringing joy that is both intense and enduring 
 D. A deep sense of joy and contentment is evident through a broad smile that reaches the eyes and a relaxed, upright posture suggesting inner peace and satisfaction with the moment 
 **Answer:** C |

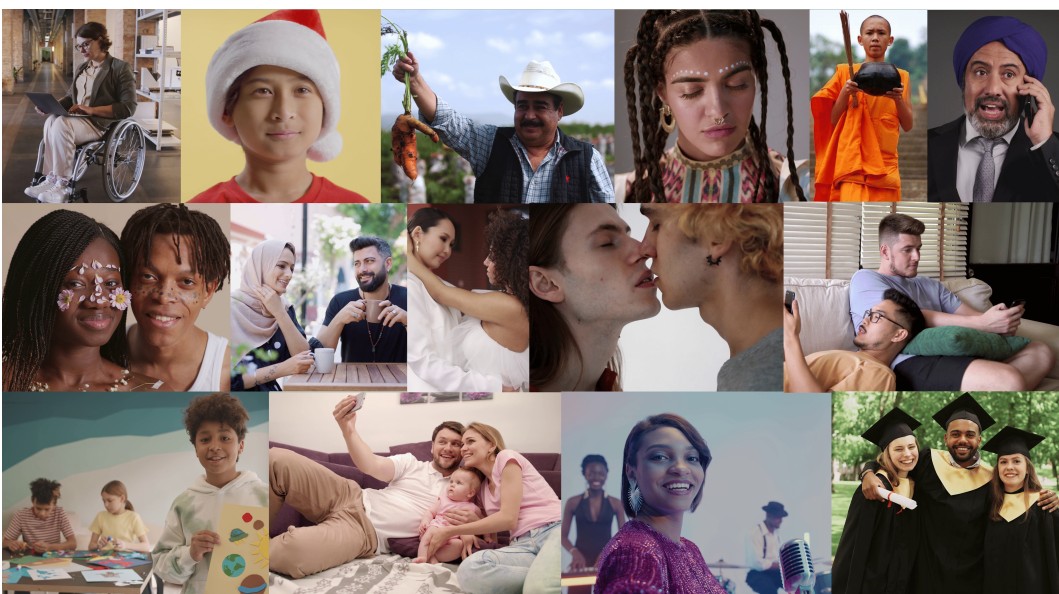

Figure 8: Samples of images used by questions in Human-MME.

# E    MODEL DETAIL

In this section, we provide detailed descriptions of the MLLMs evaluated in our experiments. Table 10 summarizes the architecture and design choices of each open-source model, including the vision encoder backbone, the underlying language model, and the total number of parameters. In addition, we indicate whether a model incorporates image grounding training or alignment, which is an important factor for interpreting their performance in different tasks.

These details complement the experimental settings described in Section 3.1. In terms of image grounding training and alignment, more specifically, GLM-4.5V (Hong et al., 2025) and GLM-4.1V-9B (Hong et al., 2025) target image grounding by aligning their outputs to a normalized xyxy-format bounding box, separated by dedicated tokens. Qwen2.5-VL (72B, 32B, 7B; (Bai et al., 2025b)) models are trained with corresponding data and restricts its outputs to JSON format. InternVL3-78B (Zhu et al., 2025) and InternVL3.5-38B (Wang et al., 2025) are also trained with image grounding data. Kimi-VL-A3B (Du et al., 2025) is trained with image grounding data but focuses mainly on GUI tasks. By examining these architectural aspects, we aim to facilitate a more transparent comparison of experimental results and highlight how design choices impact model behavior.

Table 10: Information about evaluated open-source MLLMs. Image encoding means how they map the pixels of images to tokens in language models. Grounding training means the training data and format alignment related to vision grounding.

| Model | Vision Encoder | Language Model | Image Encoding | Grounding Training |
|---|---|---|---|---|
| GLM-4.5V (106B) | AIMv2-Huge | GLM-4.5-Air | dynamic tokenization | xyxy-format bounding box with dedicated tokens |
| GLM-4.1V-9B | AIMv2-Huge | GLM-4-9B | dynamic tokenization | xyxy-format bounding box with dedicated tokens |
| Qwen2.5-VL-72B | Redesigned ViT | Qwen2.5-72B | dynamic tokenization | xyxy-format bounding box with JSON format alignment |
| Qwen2.5-VL-32B | Redesigned ViT | Qwen2.5-32B | dynamic tokenization | xyxy-format bounding box with JSON format alignment |
| Qwen2.5-VL-7B | Redesigned ViT | Qwen2.5-7B | dynamic tokenization | xyxy-format bounding box with JSON format alignment |
| InternVL3-78B | InternViT-6B | Qwen2.5-72B | 448×448 → 256 | with image grounding training data |
| InternVL3.5-38B | InternViT-6B | Qwen3-32B | 448×448 → 256 | with image grounding training data |
| InternVL3.5-241B | InternViT-6B | Qwen3-235B | 448×448 → 256 | with image grounding training data |
| Intern-S1 (241B) | InternViT-6B | Qwen3-235B | 448×448 → 256 | None |
| MiniCPM-V-4.5 (8B) | SigLIP2-400M | Qwen3-8B | 448×448 → 256 | None |
| Gemma3-27B | SigLIP-400M | Dec-only Transf. | 896×896 → 256 (whole image) | None |
| Aya-vision-32B | SigLIP2-400M | Aya Expanse 32B | 364×364 → 169 | None |
| LLaVA-NeXT-72B | CLIP-Large | Qwen1.5-72B | 336×336 → 576 | None |
| Llama-4-Scout | MetaCLIP | Llama MoE | dynamic tokenization | None |
| Kimi-VL-A3B (16B) | MoonViT | Moonlight MoE | dynamic tokenization | With GUI-focused grounding training data |
| Phi-4 (6B) | SigLIP-400M | Phi-4-Mini | 384×384 → 729 (whole image) | None |

## F  PROMPT FORMATS

For each question type, the model is guided by a structured prompt prefixed with *"Your answer should follow this format strictly:"* to ensure that the outputs are consistent and easily parseable. Table 11 summarizes the prompts used for all question types.

Table 11: Prompt formats for different question types. SC: Single-Choice, DC: Double-Choice, R: Ranking, BB: Bounding-box, SA: Short-Answer, J: Judgement.

| | | | |
|---|---|---|---|
| **SC** | Analyze: \<your analysis\> 
 Answer: A/B/C/D | **BB** | Analyze: \<your analysis\> 
 Answer: [x1,y1,x2,y2] |
| **DC** | Analyze: \<your analysis\> 
 Past: A/B/C/D 
 Future: A/B/C/D | **SA** | Analyze: \<your analysis\> 
 Answer: \<your final answer in a short and concise expression\> |
| **R** | Analyze: \<your analysis\> 
 First: A/B/C/D 
 Second: A/B/C/D 
 Third: A/B/C/D 
 Fourth: A/B/C/D | **J+SA** | Analyze: \<your analysis\> 
 Answer: \<your final answer in a short and concise expression\> 
 if no match, answer unknown |
| **J+BB** | Analyze: \<your analysis\> 
 Answer: [x1,y1,x2,y2] 
 if no match, answer [-1,-1,-1,-1] | **SA+BB** | Analyze: \<your analysis\> 
 Name: \<name of the object\> 
 Box: [x1,y1,x2,y2] |

## G  EVALUATION METRICS

We evaluate the performance of the models using metrics tailored to each type of question.

**Choice Questions**   Accuracy is used to measure correctness:

$$\text{Accuracy} = \frac{\text{Number of correct selections}}{\text{Total number of questions}}$$

**Short-Answer Questions**   We assess semantic correctness using three complementary measures:

1. BERT F1 Score (Zhang et al., 2020): compute token-level F1 between predicted answer and ground truth:

$$\text{BERT F1} = \frac{2 \cdot P_{\text{BERT}} \cdot R_{\text{BERT}}}{P_{\text{BERT}} + R_{\text{BERT}}}$$

   where $P_{\text{BERT}}$ and $R_{\text{BERT}}$ are precision and recall computed over BERT-token matches.

2. Cosine Similarity of Embeddings (Wang et al., 2020):

$$\text{CosineSim} = \frac{\mathbf{v}_{\text{pred}} \cdot \mathbf{v}_{\text{gt}}}{\|\mathbf{v}_{\text{pred}}\| \, \|\mathbf{v}_{\text{gt}}\|}$$

   where $\mathbf{v}_{\text{pred}}$ and $\mathbf{v}_{\text{gt}}$ are the embedding vectors of the predicted and ground-truth answers.

3. Keyword Coverage:

$$\text{KeywordCoverage} = \frac{|\text{Keywords}_{\text{pred}} \cap \text{Keywords}_{\text{gt}}|}{|\text{Keywords}_{\text{gt}}|}$$

The three measures are combined into a composite score to enhance robustness:

$$\text{Composite Score} = 0.5 \cdot \text{BERT F1} + 0.3 \cdot \text{CosineSim} + 0.2 \cdot \text{KeywordCoverage}$$

**Ranking Questions**  Kendall's Tau (Kendall, 1938) measures agreement between predicted and true ranking:

$$\tau = \frac{C - D}{\frac{1}{2}n(n-1)}$$

where $C$ and $D$ are the number of concordant and discordant pairs among $n$ items.

**Bounding-Box Questions**  Intersection over Union (IoU) is used:

$$\text{IoU} = \frac{\text{Area}(B_p \cap B_{gt})}{\text{Area}(B_p \cup B_{gt})}$$

where $B_p$ is the predicted bounding box and $B_{gt}$ is the ground-truth bounding box.

**Judgment Questions**  F1 score balances precision and recall:

$$\text{F1} = \frac{2 \cdot \text{Precision} \cdot \text{Recall}}{\text{Precision} + \text{Recall}}$$

This comprehensive set of metrics captures both exact correctness and semantic similarity across diverse question types.

## H FULL EXPERIMENT RESULT

Table 12: Full result of Human-MME on 20 models, 8 dimensions, 21 question types and 5 question components.

| Model | Face Understanding | | | Body Understanding | | | | HOI Understanding | | | |
|---|---|---|---|---|---|---|---|---|---|---|---|
| | Face Grounding (IoU) | Face Choice (Accuracy) | Average | Body Grounding (IoU) | Wearing Choice (Accuracy) | Wearing Short-Answer (Composite) | Average | HOI Grounding (IoU) | HOI Choice (Accuracy) | HOI Short-Answer (Composite) | Average |
| GLM-4.5V | **64.4** | 58.8 | **61.6** | **60.6** | 90.7 | 80.9 | **77.4** | **69.4** | 84.9 | **93.0** | **82.5** |
| GLM-4.1V-9B | 45.8 | 64.5 | 55.2 | 55.7 | 88.6 | 77.9 | 74.1 | 40.5 | 79.8 | 88.3 | 69.5 |
| Qwen2.5 VL-72B | 55.7 | 66.5 | 61.1 | 38.7 | 91.0 | 81.1 | 70.2 | 38.9 | 83.8 | 89.1 | 70.6 |
| Qwen2.5 VL-32B | 47.7 | 64.7 | 56.2 | 56.0 | 86.8 | 77.1 | 73.3 | 29.2 | 81.2 | 85.5 | 65.3 |
| Qwen2.5 VL-7B | 38.5 | 60.4 | 49.4 | 45.9 | 81.9 | 77.5 | 68.4 | 28.0 | 74.3 | 81.9 | 61.4 |
| Intern-S1 | 14.8 | 67.1 | 41.0 | 23.3 | 90.0 | 82.2 | 65.2 | 24.9 | 82.3 | 89.3 | 65.5 |
| InternVL3.5-241B | 32.3 | 69.2 | 50.7 | 54.2 | 90.3 | 79.4 | 74.6 | 39.0 | 84.5 | 90.7 | 71.4 |
| InternVL3-78B | 13.6 | **73.2** | 43.4 | 27.0 | **93.7** | 82.9 | 67.9 | 29.1 | 83.8 | 88.7 | 67.2 |
| InternVL3.5-38B | 20.4 | 68.8 | 44.6 | 42.4 | 93.7 | 81.7 | 72.6 | 24.3 | 81.7 | 87.8 | 64.6 |
| Llama-4-Scout | 3.0 | 51.5 | 27.3 | 11.6 | 69.8 | 70.4 | 50.6 | 8.3 | 61.4 | 78.3 | 49.4 |
| LLaVA-NeXT-72B | 13.2 | 62.8 | 38.0 | 38.4 | 85.2 | 76.9 | 66.8 | 35.8 | 78.4 | 81.0 | 65.1 |
| Aya-vision-32B | 6.4 | 55.5 | 30.9 | 13.5 | 75.9 | 82.2 | 57.2 | 10.2 | 77.0 | 84.3 | 57.1 |
| Gemma3-27B | 8.0 | 62.1 | 35.1 | 16.1 | 83.1 | 80.5 | 59.9 | 16.3 | 81.2 | 86.2 | 61.2 |
| Kimi-VL-A3B | 15.2 | 59.4 | 37.3 | 29.7 | 83.3 | 76.2 | 63.1 | 11.1 | 75.3 | 65.9 | 50.8 |
| MiniCPM-V-4.5 | 12.3 | 65.5 | 38.9 | 24.5 | 88.9 | 74.4 | 62.6 | 17.3 | 82.5 | 87.5 | 62.4 |
| Phi-4 | 6.0 | 53.0 | 29.5 | 12.1 | 59.2 | 72.9 | 48.1 | 8.5 | 72.3 | 64.9 | 48.6 |
| Gemini-2.5-Pro | 17.1 | 67.6 | 42.4 | 27.5 | 89.0 | **83.0** | 66.5 | 31.6 | **87.1** | 91.2 | 70.0 |
| GPT-4o | 5.7 | 52.0 | 28.8 | 13.1 | 82.8 | 80.6 | 58.8 | 18.0 | 76.4 | 85.1 | 59.8 |
| GPT-5 | 14.7 | 54.2 | 34.4 | 32.0 | 91.2 | 80.4 | 67.8 | 37.2 | 84.9 | 91.1 | 71.1 |
| GPT-5-mini | 10.4 | 50.8 | 30.6 | 25.4 | 91.0 | 82.6 | 66.3 | 29.8 | 85.4 | 87.1 | 67.4 |

| Model | Multi-Person Reasoning | | | | | | | | | | |
|---|---|---|---|---|---|---|---|---|---|---|---|
| | Identify Grounding (IoU) | Identify Choice (Accuracy) | Identify Short-Answer (Composite) | Identify Open HOI (Composite) | Identify Open HOI (IoU) | Judgment Grounding (F1) | Judgment Grounding (IoU) | Judgment Short-Answer (F1) | Judgment Short-Answer (Composite) | Common Choice (Accuracy) | Average |
| GLM-4.5V | **65.8** | 62.3 | 81.6 | 85.8 | **72.6** | 63.8 | | 68.3 | 80.1 | 71.5 | **71.5** |
| GLM-4.1V-9B | 45.0 | 59.9 | 79.1 | 84.0 | 66.5 | 69.2 | 46.8 | 63.4 | 76.9 | 62.9 | 64.3 |
| Qwen2.5 VL-72B | 57.3 | **64.9** | 79.8 | 81.9 | 61.4 | 67.5 | **49.7** | **75.2** | 80.7 | 46.2 | 65.2 |
| Qwen2.5 VL-32B | 46.4 | 61.8 | 74.3 | 65.7 | 47.0 | 65.3 | 43.2 | 66.6 | 66.1 | 46.9 | 58.2 |
| Qwen2.5 VL-7B | 33.2 | 47.3 | 73.8 | 56.6 | 17.1 | 56.5 | 23.5 | 56.4 | 69.4 | 29.9 | 46.3 |
| Intern-S1 | 22.6 | 64.1 | 79.3 | 86.2 | 26.6 | **70.3** | 19.9 | 67.5 | 78.5 | 74.3 | 59.3 |
| InternVL3.5-241B | 47.2 | 58.1 | 79.1 | 82.9 | 61.8 | 58.2 | 44.5 | 56.5 | 79.0 | 40.1 | 59.4 |
| InternVL3-78B | 28.2 | 59.8 | 79.4 | **86.9** | 35.8 | 62.6 | 25.8 | 60.7 | **81.9** | 38.2 | 54.6 |
| InternVL3.5-38B | 28.7 | 57.4 | 77.9 | 83.8 | 42.4 | 66.0 | 25.8 | 58.0 | 78.2 | 35.2 | 53.8 |
| Llama-4-Scout | 6.6 | 31.7 | 67.9 | 74.0 | 5.1 | 28.9 | 1.8 | 48.3 | 62.4 | 21.1 | 33.9 |
| LLaVA-NeXT-72B | 27.2 | 43.1 | 68.8 | 75.0 | 39.6 | 53.0 | 24.8 | 51.7 | 71.1 | 34.0 | 47.2 |
| Aya-vision-32B | 8.7 | 43.1 | 75.8 | 82.7 | 7.6 | 55.7 | 6.3 | 51.9 | 74.4 | 32.9 | 42.8 |
| Gemma3-27B | 15.5 | 39.8 | 77.0 | 75.9 | 11.8 | 55.4 | 15.3 | 53.7 | 77.7 | 38.2 | 45.1 |
| Kimi-VL-A3B | 19.5 | 40.2 | 73.1 | 72.2 | 9.0 | 46.2 | 19.6 | 54.7 | 72.2 | 28.8 | 42.6 |
| MiniCPM-V-4.5 | 24.5 | 49.3 | 79.4 | 84.4 | 21.3 | 58.9 | 20.5 | 57.0 | 78.6 | 50.9 | 52.1 |
| Phi-4 | 4.1 | 32.3 | 66.9 | 49.5 | 0.2 | 7.0 | 0.7 | 32.7 | 59.9 | 29.0 | 29.6 |
| Gemini-2.5-Pro | 23.8 | 56.9 | **82.2** | 86.4 | 20.0 | 69.9 | 19.1 | 74.1 | 80.4 | **74.6** | 58.9 |
| GPT-4o | 10.8 | 38.7 | 75.9 | 85.9 | 12.8 | 47.7 | 6.9 | 49.5 | 70.6 | 27.4 | 41.4 |
| GPT-5 | 24.3 | 53.0 | 74.0 | 86.7 | 27.1 | 38.2 | 9.6 | 45.2 | 65.6 | 55.6 | 49.0 |
| GPT-5-mini | 21.7 | 51.5 | 71.9 | 86.7 | 23.7 | 47.1 | 11.5 | 51.2 | 68.0 | 48.8 | 48.3 |

| Model | Multi-Image Understanding | | | | Intention Discrimination | Cause Discrimination | | | Emotion Discrimination |
|---|---|---|---|---|---|---|---|---|---|
| | Multi-Face (Tau) | Multi-Wearing (Tau) | Multi-HOI (Accuracy) | Average | Intention Choice (Accuracy) | Causal Choice Accuracy (Past) | Causal Choice Accuracy (Future) | Average | Emotion Choice (Accuracy) |
| GLM-4.5V | 86.1 | 86.1 | 65.4 | 79.2 | 83.9 | 85.6 | 85.1 | 85.4 | 66.6 |
| GLM-4.1V-9B | 76.8 | 76.7 | 62.0 | 71.8 | 82.7 | 75.3 | 76.8 | 76.0 | 58.8 |
| Qwen2.5 VL-72B | 81.5 | 84.1 | 60.6 | 75.4 | **88.1** | 85.4 | 87.2 | 86.3 | 65.3 |
| Qwen2.5 VL-32B | 75.3 | 78.5 | 58.3 | 70.7 | 82.9 | 80.3 | 81.8 | 81.1 | 64.9 |
| Qwen2.5 VL-7B | 63.4 | 65.8 | 53.7 | 61.0 | 84.1 | 67.4 | 76.8 | 72.1 | 60.9 |
| Intern-S1 | 85.1 | 87.2 | 67.1 | 79.8 | 82.9 | 83.1 | 83.3 | 83.2 | **68.3** |
| InternVL3.5-241B | 83.0 | 83.9 | 62.1 | 76.4 | 83.7 | 77.8 | 87.2 | 82.5 | 66.4 |
| InternVL3-78B | 84.9 | 85.4 | 65.5 | 78.6 | 86.7 | 84.7 | 84.7 | 84.7 | 67.7 |
| InternVL3.5-38B | 78.0 | 84.2 | 62.8 | 75.0 | 86.9 | 79.1 | 77.0 | 78.0 | 65.6 |
| Llama-4-Scout | 51.1 | 51.3 | 44.2 | 48.9 | 66.5 | 54.4 | 59.8 | 57.1 | 50.4 |
| LLaVA-NeXT-72B | 60.9 | 60.1 | 43.3 | 54.8 | 77.0 | 68.4 | 72.6 | 70.5 | 54.6 |
| Aya-vision-32B | 78.3 | 73.0 | 52.3 | 67.9 | 76.2 | 68.6 | 74.9 | 71.8 | 57.4 |
| Gemma3-27B | 75.7 | 72.8 | 47.4 | 65.3 | 81.5 | 69.0 | 77.0 | 73.0 | 60.1 |
| Kimi-VL-A3B | 23.3 | 7.6 | 51.0 | 27.3 | 81.0 | 65.5 | 60.7 | 63.1 | 55.3 |
| MiniCPM-V-4.5 | 78.8 | 80.1 | 61.7 | 73.5 | 81.5 | 68.8 | 66.7 | 67.8 | 63.3 |
| Phi-4 | 45.6 | 42.5 | 30.6 | 39.6 | 62.9 | 41.4 | 34.7 | 38.1 | 46.4 |
| Gemini-2.5-Pro | **88.1** | **91.1** | **71.5** | **83.6** | 79.4 | 87.4 | 84.7 | 86.1 | 64.5 |
| GPT-4o | 77.9 | 85.5 | 60.7 | 74.7 | 79.2 | 74.3 | 78.0 | 76.2 | 52.7 |
| GPT-5 | 73.0 | 89.0 | 65.5 | 75.8 | 82.3 | **90.4** | 88.1 | **89.2** | 42.6 |
| GPT-5-mini | 73.7 | 89.6 | 65.9 | 76.4 | 79.4 | 82.6 | 80.8 | 81.7 | 39.9 |

# I DETAILED DISCUSSION AND FINDINGS

Here we provide detailed discussion for findings in Section 3.3.

**Stronger scaling effects in human related Choice and Ranking tasks.** According to Figure 4, performance on Choice and Ranking question components have the strongest correlation to model size. The significant positive correlations can be attributed to the fact that models with larger parameter counts are better able to attend to and integrate a greater number of visual features simultaneously. In Choice tasks this capability allows the model to evaluate multiple candidate options in parallel and distinguish subtle differences among them. For Ranking, which involves reasoning across multiple images and combining numerous facial or clothing attributes, the ability to consider many features at once is even more critical. As a result, increasing model size directly enhances performance in these settings, leading to stronger correlations compared with other metrics.

**Training data have a strong influence on human related grounding task.** In Figure 4, the performance on bounding box tasks shows almost no correlation with model size. We therefore examine a range of factors that could potentially influence grounding capability, including the choice of vision encoder, the language model backbone, the method used to convert images into tokens, and the presence of grounding-relevant training data and format alignment. Our analysis indicates that training data and output-format alignment provide the most substantial benefits for visual grounding. A detailed comparison of model architectures is provided in Table 10, and the corresponding bounding box results appear in Table 2.

MiniCPM-V-4.5 and Aya-vision-32B both employ the same SigLIP2-400M (Tschannen et al., 2025) vision encoder and neither reports any specialized training on visual grounding tasks. Despite having far fewer parameters, MiniCPM-V-4.5 achieves a clearly higher IoU than Aya-vision-32B. Gemma3-27B and Phi-4 also share a common vision encoder, SigLIP-400M (Zhai et al., 2023), yet their grounding performance differs substantially. Within the GLM, Qwen, and Intern families, models that use identical vision encoders still display large within-family variation. These observations indicate that the vision encoder is not the dominant factor affecting grounding outcomes.

Regarding the language model, MiniCPM-V-4.5 uses Qwen3-8B, which has stronger general capability than the Qwen2.5-7B backbone used by Qwen2.5-VL-7B. Nonetheless, MiniCPM-V-4.5 performs worse on grounding than Qwen2.5-VL-7B. Similarly, Intern-S1 adopts the largest Qwen3-235B language model in the comparison, yet its grounding performance is weaker than that of Qwen2.5-VL-32B, which relies on the smaller previous-generation Qwen2.5-7B. These results indicate that the language model itself has limited influence on bounding box performance.

In terms of image tokenization, the best-performing GLM and Qwen models use dynamic tokenization rather than mapping fixed-resolution image tiles to fixed token counts. However, Kimi-VL-A3B adopts a similar mechanism yet performs markedly worse than all GLM and Qwen models on grounding tasks. Substantial performance differences can also be observed among models that use the same image-token conversion strategy, showing no consistent trend.

The picture changes once training data are considered. The models with the strongest performance, namely GLM, Qwen, and InternVL, all explicitly report the inclusion of visual grounding data in their training corpus. Although Kimi-VL-A3B also mentions grounding-related data, its training is restricted to OS-agent grounding, whose domain differs substantially from the human-centric imagery used in our benchmark.

A further piece of compelling evidence comes from comparing Intern-S1 with InternVL3.5-241B. Apart from differences in training data and the special tokens introduced in Intern-S1 to enhance its scientific capabilities, the two models share an almost identical architecture. Nevertheless, their performance on human-centric visual grounding diverges substantially, with Intern-S1 performing markedly worse. This gap is consistent with the fact that Intern-S1 is designed as a science-oriented large model whose training data are correspondingly biased, and its technical report does not explicitly mention training on visual grounding tasks.

Moreover, among models of comparable scale, grounding performance follows a consistent pattern: GLM outperforms Qwen, which in turn outperforms InternVL. This ordering matches the degree of format alignment described in their technical reports. InternVL does not mention any dedicated bounding box format alignment. Qwen states that it aligns bounding box outputs to JSON

with absolute xyxy coordinates. GLM constrains outputs to normalized xyxy coordinates and introduces dedicated start and end tokens for bounding boxes. The progressively stronger alignment requirements correspond directly to progressively stronger grounding performance, reinforcing the conclusion that both training data and output-format alignment play a decisive role in achieving high-quality visual grounding.

**Challenges in left-right discrimination for body parts.** Figure 5 presents confusion matrices for three MLLMs of different architectures and parameter scales on the Face Grounding and Body Grounding tasks. Across all six matrices, it is evident that these models encounter notable difficulty in distinguishing between the left and right hands or feet, whereas their ability to differentiate left from right on facial features shows significantly stronger. Even Qwen2.5-VL-72B, which overall appears relatively robust, exhibits a clear tendency toward such confusion. A plausible explanation is that the left-right configuration of facial components remains fixed in image space: when a person faces away from the camera the face is simply not visible. It is impossible for a person's real left eye to appear on the opposite side of the nose, unless we could somehow see their eye through the back of their head. In contrast, the human body does not impose such a constraint, and the left and right hands may appear on either side of the torso depending on pose or viewpoint. This difference during both training and testing makes left-right discrimination of body parts more challenging for the models.

**Judgment tasks have precision-recall tradeoff.** Table 3 reports the precision, recall, and F1 score of different models when deciding whether to answer or abstain on Judgment-type questions. Most models exhibit relatively low precision, indicating that they often fail to abstain when no suitable person is present in the image. This reflects a persistent tendency toward hallucination. By contrast, recall is generally high, showing that when the correct individual is indeed present, the models usually succeed in identifying the target.

Among all models, Qwen2.5-VL-72B achieves the highest overall F1 score, but its behavior reveals a notable tradeoff. Its recall is slightly lower than that of Qwen2.5-VL-32B, suggesting that its stronger control over hallucination comes at the cost of missing some valid answers. A case-by-case inspection further highlights this effect: even on Body Grounding tasks that contain no judgment component, Qwen2.5-VL-72B frequently refuses to output the full body bounding box, citing incomplete visibility of the human body despite explicit instructions to annotate "all visible body". As a result, its final IoU for body drops to only 0.31, far below the 0.89 achieved by Qwen2.5-VL-7B. By examining the pattern of refusal responses, we find that it is likely a byproduct of CoT alignment. This suggests that during the collection of CoT samples, the dataset may have overrepresented answers to questions about objects that do not exist in the image. As a result, the model sometimes refuses to answer a legitimate question by incorrectly claiming that "the requested object does not exist", particularly in certain specific scenarios. These observations suggest that models with stronger hallucination prevention mechanisms may, paradoxically, follow instructions less faithfully because they act with excessive caution.

**Extra Judgment question component reduces performance on original task.** Table 3 also presents the effect of adding a Judgment component on related tasks. The four columns on the right compare model performance on Identify Bounding Box versus Judgment Bounding Box, and Identify Short-Answer versus Judgment Short-Answer. In Identify tasks, the model must locate the correct individual based on a single distinguishing feature and then answer the question. In Judgment tasks, MLLMs must first identify a person satisfying two specified features and, only when such a person exists, provide the answer. Across most models and metrics, the Judgment versions yield lower performance than their Identify counterparts, indicating that the added complexity of dual-feature matching makes final question completion more challenging, although the extra feature provides additional cues for localization.

**Intention discrimination is easier than cause discrimination, which in turn is easier than emotion discrimination.** Table 2 shows that, across all evaluated models, accuracy almost consistently follows the pattern **Intention > Cause > Emotion**. Intention discrimination benefits from strong, visually grounded cues such as body posture, gaze direction, and surrounding objects, making it comparatively straightforward for models to infer a person's likely goal or purpose. Cause discrimination requires imagining scene-level causes and predicting future consequences, which depend on higher-level commonsense reasoning and temporal context that are not directly observable, thereby lowering accuracy. Emotion discrimination proves to be the most challenging because emotional

states are inherently subtle and subjective, facial expressions can be ambiguous or culturally variable. This progressive increase in abstraction explains the observed hierarchy of model performance.

**Poor performance of proprietary models on certain tasks.** We observe that among the four proprietary models tested, even the strongest ones, such as Gemini-2.5-Pro and GPT-5, still fall behind the best-performing open-source model on certain metrics. This pattern can be explained by three factors: the safety alignment applied to commercial models, inherent limitations in their grounding abilities, and the general performance constraints of older model architectures. Table 4 reports the proportion of refusal cases across eight evaluation dimensions. We identify a refusal by checking whether the output contains the word "sorry". Only GPT-5 and GPT-5-mini exhibit such behaviors, which matches the emphasis on safety alignment described in their system card (OpenAI, 2025).

A closer, case-by-case examination shows that GPT-5 often refuses to provide detailed analysis of facial attributes because its safe-completion mechanism is triggered. In these cases, the model tends to abandon the original instructions or required format and instead proposes an alternative task, which is usually irrelevant or inappropriate. Table 5 shows the frequency with which these refusals mention keywords related to policy or privacy (keywords: "policy", "policies", "prohibit", "sensitive", "not appropriate", "privacy"). These instances are mostly caused by tasks that require inference about gender, race, or other appearance-related features. The most affected dimensions include Face Understanding, Emotion Discrimination, and other categories that involve facial information are also affected. This behavior appears only in the GPT-5 series; GPT-4o and all other open-source or proprietary models do not show similar patterns. The overly conservative alignment practices in GPT-5 therefore have a clear negative impact on its evaluation scores.

Even after removing the influence of safety alignment for GPT-5, and even for models not affected by such constraints, such as Gemini-2.5-Pro or GPT-4o, we still observe a considerable performance gap relative to the leading open-source model. As shown in Table 12, this gap is concentrated in IoU metrics, which correspond to bounding-box tasks. We further break down IoU performance by bounding-box type and compare each proprietary model with the top open-source model, as shown in Table 6. A consistent pattern emerges. All proprietary models perform worse than GLM-4.5V overall, but the degree of underperformance varies. For coarse-grained boxes such as whole face and whole body, the decline is moderate, roughly a 50% reduction, indicating that these models can still produce global annotations on images. For fine-grained body parts, however, the performance drop is dramatic, often falling to one-fifth or even one-hundredth of the open-source baseline. For comparison, another relatively weaker open-source model, InternVL3.5-38B, shows similar performance to the proprietary models on coarse-grained boxes, but remains significantly stronger than all proprietary models on fine-grained facial and body-part annotations. This suggests a consistent cognitive bias in the proprietary models when dealing with human-image grounding. They tend to recognize entire faces or bodies clearly, but their understanding of specific body parts is much less accurate. This pattern appears consistently across all four proprietary models we evaluated.

The most plausible explanation is the imbalance in training data. Open-source datasets often contain many annotations for whole faces and whole bodies, while annotations for individual body parts are much rarer, which limits the models' ability to generalize to fine-grained grounding tasks. This gap may stem from the slower iteration pace of closed-source models, coupled with the fact that the requirements for fine-grained visual alignment in semantic tasks tend to favor scenarios where open-source models perform better than closed-source models. Common scenarios for fine-grained visual alignment in semantic tasks include large-scale data annotation and fine-tuning for downstream tasks. Both of the scenarios are more conducive to open-source models than closed-source ones. Consequently, the latest open-source models have also been optimized for the category of task accordingly.

Among the proprietary models, GPT-4o performs notably worse than the others. This observation aligns with findings reported in several previous studies (Fu et al., 2024b; Lu et al., 2024; Zhang et al., 2024b). GPT-4o is an older model released in mid-2024, and its weaker performance relative to more recently released systems, including open-source models, is therefore expected.

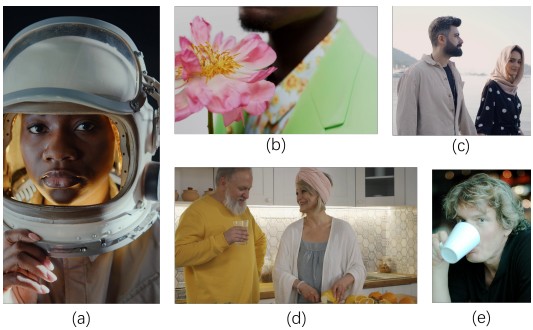

Figure 9: Images for case study on failures of best models.

## J    FURTHER CASE STUDY AND FINDINGS

By examining the errors made by state-of-the-art models (GLM-4.5V and Gemini-2.5-Pro) across different tasks, we identify several shortcomings that these models still exhibit. For example, GLM-4.5V tends to commit to an answer prematurely and then craft its reasoning around that predetermined conclusion, rather than analyzing first and deciding afterward. Due to the autoregressive nature of LLMs, analysis should logically precede the final answer to achieve higher accuracy; however, current models sometimes output an answer before conducting a full evaluation. In the task corresponding to Figure 9(a), which asks the model to select the most accurate facial feature, GLM-4.5V gives the following reasoning: "Looking at the image, the person's face has a rounded and full appearance, which matches the description of a chubby face. The other options (big nose, narrow eyes, bushy eyebrows) do not accurately describe the facial features visible here". Here, the model implicitly locks onto the "chubby face" option before performing a comprehensive comparison. In contrast, a step-by-step analysis followed by a final decision often produces better results; Gemini-2.5-Pro adopts this strategy on the same question and successfully reaches the correct answer.

GLM-4.5V shows limited discrimination ability for fine-grained clothing categories. In Figure 9(b), the model is asked to choose the option consistent with the person's clothing. It is misled by a distractor describing a "light green fluffy jacket". Although the individual does wear a jacket-like garment, it is not a fluffy jacket. The correct option should be "a white and yellow shirt". The jacket is more visually salient but mismatched in category; the less salient inner shirt, which exactly matches one option, is overlooked. A similar issue appears in Figure 9(c), where the model must identify a shared attribute of two people. One distractor claims both individuals are "wearing black shirts," yet the woman is clearly wearing a tunic or dress rather than a shirt, so this option should be eliminated. The correct answer is that neither person has a double chin, which is visually apparent.

Despite its strong ability to predict bounding boxes, GLM-4.5V is less capable of inferring object identity directly from coordinates. In Figure 9(d), the model is asked whether a bearded person appears within the bounding box [961,96,1129,355] in a 1920×1080 image. GLM-4.5V incorrectly answers that such a person exists. Its reasoning shows that once it concludes the box lies "near the center-left", it immediately assumes the region corresponds to the man on the left, even though the box is not actually on the left and the person inside is a woman, not a bearded man.

Gemini-2.5-Pro also makes orientation-related mistakes, particularly in distinguishing left from right when analyzing facial direction. Since closed-source models generally perform poorly on grounding tasks, it is difficult to rely on IoU metrics to verify whether they truly understand left–right distinctions; nonetheless, other examples also suggest such issues. In a Multi-Face task requiring the model to rank face images by the number of attributes they satisfy, one attribute is "face turned to the left side of the image". In the example shown in Figure 9(e), both the viewing angle and facial cues clearly indicate that the face is turned to the left, yet Gemini-2.5-Pro incorrectly states that the face is turned to the right, which leads to an incorrect count of satisfied attributes.

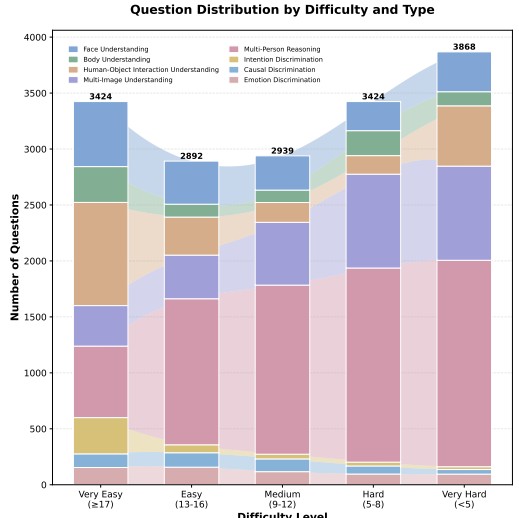
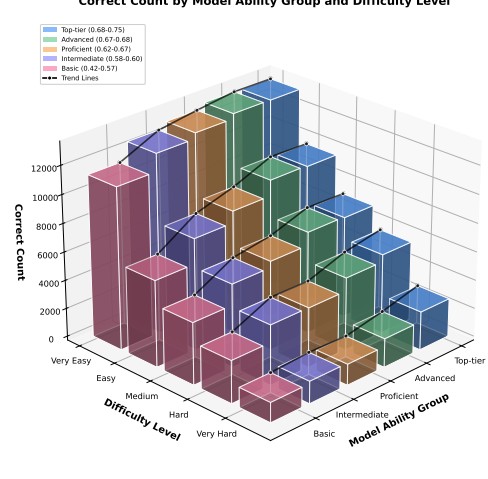

Figure 10: **Difficulty analysis of 19,945 questions in Human-MME.** The questions are categorized into five levels based on their accuracy rates. Models are grouped into five tiers according to their average scores across all dimensions. The figure presents the difficulty distribution of questions across eight different dimensions, as well as how the total number of correct answers changes for models of different capability tiers when facing questions of varying difficulty.

## K  DIFFICULTY ANALYSIS

We aggregate the responses of 20 models for each question and categorize the questions into five groups based on the number of models that answered them correctly: very easy (17–20 models correct), easy (13–16 models correct), medium (9–12 models correct), hard (5–8 models correct), and very hard (fewer than 5 models correct). The stacked bar chart on the left side of Figure 10 shows the difficulty distribution both overall and across each individual dimension. Although different dimensions manifest distinct difficulty tendencies, once they are combined into the full Human-MME dataset, the diversity of question types results in a remarkably balanced difficulty distribution. The difference between the largest and smallest question counts across difficulty levels is within 1000, ensuring sufficient discriminative power for evaluating models.

We then rank the models according to their average scores and grouped them in sets of four, dividing the 20 MLLMs into five tiers: Basic, Intermediate, Proficient, Advanced, and Top-tier. As shown in the bar chart on the right side of Figure 10, for each difficulty level, the number of correctly answered questions steadily increases as model performance improves. Moreover, the easier the questions are, the more substantial the improvement among weaker models; conversely, the harder the questions are, the larger the performance gap among stronger models. Within each model tier, we observe a consistent pattern in which easier questions lead to a higher number of correct answers. Taken together, these observations demonstrate that Human-MME is a benchmark with a balanced and fair difficulty distribution, providing meaningful discrimination across models of all capability levels.

## L  THE USE OF LARGE LANGUAGE MODELS

We use large language models solely for polishing our writing, and we have conducted a careful check, taking full responsibility for all content in this work.

