# OpenReview forum: "Human-MME: A Holistic Evaluation Benchmark for Human-Centric Multimodal Large Language Models"
_ICLR.cc/2026/Conference — ICLR 2026 Poster_

### Official Review · Reviewer_ZsgC · 2025-10-29

**Soundness:** 3
**Presentation:** 3
**Contribution:** 3
**Rating:** 4
**Confidence:** 4

**Summary:**

This paper proposes Human-MME, a comprehensive evaluation benchmark for the human-centric scene understanding capabilities of Multimodal Large Language Models (MLLMs). It aims to address the issue that existing benchmarks lack evaluations of fine-grained human perception and high-dimensional causal reasoning. The benchmark secures the quality of annotations by collecting and screening 16,765 high-quality images from free stock image libraries and the HICO-DET dataset, combined with an "automated annotation (5-step process) + manual adjustment (cluster deduplication and instance correction)" approach. It designs 19,945 image-question pairs covering 8 evaluation dimensions (such as face understanding and causal discrimination) and 5 types of question-answer paradigms. Experiments on 17 mainstream MLLMs show that GLM-4.5V achieves the best overall performance, while Qwen2.5-VL-72B excels in high-level reasoning. Meanwhile, the experiments reveal the limitations of the models in aspects like left-right discrimination of body parts and precision-recall trade-off in judgment tasks, providing support for research on human-centric scene understanding of MLLMs.

**Strengths:**

- To address the issue that existing benchmarks lack comprehensive evaluation of "fine-grained perception + high-dimensional reasoning" in human-centric scenarios, this study constructs, for the first time, an 8-dimensional progressive evaluation system ranging from face/body understanding (fine-grained) to intention/causality/emotion discrimination (high-level reasoning). This system covers the core capability dimensions of human scene understanding and can more comprehensively reveal the capability boundaries of Multimodal Large Language Models (MLLMs) in this field.
- Five types of question-answer paradigms are designed, including multiple-choice questions, bounding box questions, short-answer questions, ranking questions, and judgment questions, to adapt to different evaluation needs. Moreover, each type of paradigm is matched with exclusive and precise metrics (e.g., Intersection over Union (IoU) for bounding box questions, Kendall’s τ for ranking questions, and F1 score for judgment questions). This avoids the flaw of a "one-size-fits-all" metric approach and enables a more objective assessment of model performance across different tasks.

**Weaknesses:**

- The annotation of high-level reasoning tasks such as intention, emotion, and causality relies on Qwen3 to generate "identity-neutral descriptions". Although manual verification is conducted, there are still certain subjective differences in judgments like "reasonableness of intention" and "emotion matching degree" (for example, different experts may have different opinions on whether "a person's intention is 'preparing for an adventure' or 'organizing equipment'"), which may affect the objectivity of the evaluation for such tasks.
- The human-centric scenes in the dataset do not explicitly mention cross-cultural diversity (such as differences in clothing and behavioral habits across regions), nor are there specially designed evaluation scenarios for specific populations (such as people with disabilities and groups of different age groups). This makes it difficult for the benchmark to measure issues like "cultural bias" or "insufficient population coverage" of MLLMs in human-centric scene understanding.

**Questions:**

It is suggested that the authors add a table to comparatively demonstrate the differences and advantages between Human-MME and previous relevant benchmarks.

---

> ### Author Response · Authors · 2025-11-24
>
> # Rebuttal
>
> We sincerely thank the reviewers for their thoughtful comments and constructive suggestions. We address the raised concerns point-by-point below.
>
> ---
>
> > **Q1: On the subjectivity of high-level reasoning tasks (intention, emotion, causality)**
>
> We agree that intention, emotion, and causality involve inherently subjective components. However, this subjectivity does *not* compromise the objectivity of Human-MME as an evaluation instrument. In designing the benchmark, we consider this issue from the start:
>
> 1. **Question Format Design to Avoid Subjectivity Bias**
>
>   For high-level reasoning dimensions (intention, emotion, causality), we intentionally avoid using ranking and short-answer questions, which can be sensitive to subjective phrasing or model-specific writing styles. Instead, these three dimensions are evaluated **exclusively with multiple-choice questions (choose one)**, whose scoring objectivity is well established in **Classical Test Theory (CTT)**. Even when multiple reasonable options exist, as long as our standard answers are stable, have objective scoring rules, and errors are random, the subjectivity within the options does not compromise the overall objectivity of the evaluation.
>
>   Furthermore, the ground truth in our benchmark does not rely on the textual content generated by Qwen3. Instead, what we rely on is the fact that the text is generated based on the image. The correct option comes form the same image while the distractors comes from similar but different images. There is a natural correspondence between the image and the correct option. In our evaluation, the model's task is to infer the emotions, intentions, and other attributes in the text from the image and to recover this inherent correspondence.
>
> 2. **Reliability Validation via Inter-Expert Agreement**
>
>  To further confirm the objectivity and consistency of our annotations, we introduce the Fleiss' κ metric to validate agreement among humans. **Fleiss' κ** (Fleiss, 1971) is a standard statistical measure that evaluates the agreement among annotation sources. For a single-choice question with four options, it ranges form –0.33 (always disagree with each other) to 1 (identical). We randomly select 10% of questions from Intention Discrimination and Emotion Discrimination and ask 6 volunteers to make their own choice. We then compute Fleiss' κ among 6 volunteers, and get 0.660. The value suggests substantial agreement among experts, which demonstrates that the subjective preferences are subtle.
>
> ---
>
> > **Q2: On cultural bias and population diversity**
>
> We appreciate the reviewer's insightful comment regarding cultural bias and population coverage. In Human-MME:
>
> 1. **Diverse Human-Centric Imagery**
>    The dataset already includes images covering **a wide range of cultural, ethnic, age, and lifestyle backgrounds**. Additional representative samples are provided in **Appendix D (Figure 8)**.
> 2. **Scope Definition of Our Benchmark**
>    Human-centric scene understanding is a broad research domain. Our benchmark focuses on *fine-grained human understanding and foundational perception-to-reasoning capabilities*.
>    Dimensions such as cultural bias and population-specific performance, while highly important, represent a *different research axis* and require dedicated evaluation protocols.
> 3. **Dataset Release for Future Research**
>    Importantly, we will publicly release the **complete dataset containing 16,765 images**, along with all annotations in the camera-ready version. This provides a strong foundation for future work on cultural bias, fairness, and inclusiveness.
>
> ---

---

> > ### Author Response · Authors · 2025-11-24
> >
> > > **Q3: Addition of comparison tables**
> >
> > We thank the reviewer for the suggestion. Following the advice, we enhance the original **Table 1** for clearer comparison, and add a new **Table 7** in **Appendix A** to compare Human-MME with existing benchmarks from a *methodological perspective*.
> >
> > These tables highlight the key advantages of Human-MME in terms of comprehensiveness, fine-grained evaluation, annotation pipeline, and quality control.
> >
> > For convenience, we include the updated tables below in the rebuttal.
> >
> > | Benchmark            | Modality    | #QA  | Formats                                   | Fine-grained grounding | Face Features | Body Features | Human-Object Interaction | Multi-image | Multi-person | High-level abstract features |
> > | -------------------- | ----------- | ---- | ----------------------------------------- | ---------------------- | ------------- | ------------- | ------------------------ | ----------- | ------------ | ---------------------------- |
> > | MMBench              | Image       | 3.2K | Choice                                    | ✗                      | ✓             | ✗             | ✓                        | ✗           | ✓            | ✓                            |
> > | MME                  | Image       | 2.8K | TF                                        | ✗                      | ✗             | ✓             | ✗                        | ✗           | ✓            | ✗                            |
> > | Seed-Bench           | Image+Video | 19K  | Choice                                    | ✗                      | ✗             | ✗             | ✓                        | ✗           | ✓            | ✗                            |
> > | HV-MMBench           | Video       | 8.7K | Choice / Open-ended / TF                  | ✗                      | ✗             | ✗             | ✗                        | ✗           | ✓            | ✓                            |
> > | HumanVBench          | Video       | 2.1K | Choice                                    | ✗                      | ✗             | ✓             | ✗                        | ✗           | ✓            | ✓                            |
> > | Face-Human-Bench     | Image       | 2.7K | Choice                                    | ✗                      | ✓             | ✗             | ✗                        | ✓           | ✓            | ✓                            |
> > | HumaniBench          | Image       | 32K  | Choice / Open-ended / BBox                | ✗                      | ✗             | ✓             | ✓                        | ✗           | ✓            | ✓                            |
> > | **Human-MME (Ours)** | Image       | 20K  | Choice / Open-ended / TF / BBox / Ranking | ✓                      | ✓             | ✓             | ✓                        | ✓           | ✓            | ✓                            |
> >
> > | Benchmark            | Scale   | Data Source                | Labeling                              | QA Curation | Quality Assurance                             |
> > | -------------------- | ------- | -------------------------- | ------------------------------------- | ----------- | --------------------------------------------- |
> > | MMBench              | 3.2K    | Public datasets & Internet | No labeling; QAs directly constructed | Human       | Human verification after multiple LLM filters |
> > | MME                  | 2.8K    | Public datasets            | No labeling; QAs directly constructed | Human       | Not mentioned                                 |
> > | Seed-Bench           | 19K     | Public datasets            | Dataset captions & FM-generated text  | GPT-4       | Human verification                            |
> > | HV-MMBench           | 8.7K    | Internet                   | Qwen2.5                               | Qwen2.5     | Human verification after Qwen2.5 filtering    |
> > | HumanVBench          | 2.1K    | Internet                   | Video-MLLM generated                  | LLMs        | Human verification                            |
> > | Face-Human-Bench     | 2.7K    | Public datasets            | Dataset captions                      | Templates   | Human verification                            |
> > | HumaniBench          | 32K     | Internet                   | GPT-4o                                | Templates   | Human verification                            |
> > | **Human-MME (Ours)** | **20K** | Public datasets + Internet | **Specialized expert models + LLMs**  | Templates   | **Human filtering + correction**              |
> >
> > Citations to the above papers have been incorporated into our paper.
> >
> > ## **References**
> >
> > Fleiss, 1971. Measuring nominal scale agreement among many raters. Psychological Bulletin, 1971.
> >
> > ---
> >
> > We appreciate the reviewers' positive recognition of our benchmark’s contribution. Revisions based on your feedback further strengthen the clarity, rigor, and completeness of our work. Thank you again for your valuable feedback.

---

### Official Review · Reviewer_rEtP · 2025-10-31

**Soundness:** 3
**Presentation:** 3
**Contribution:** 3
**Rating:** 8
**Confidence:** 4

**Summary:**

This paper addresses the gap of insufficient human-centric evaluation benchmarks for Multimodal Large Language Models (MLLMs) by proposing **Human-MME**, a holistic benchmark for human-centric scene understanding. Key features of Human-MME include: (1) Diverse human scenes (4 primary domains, 15 secondary domains, 43 sub-fields) and 19,945 real-world image-question pairs; (2) Progressive evaluation across 8 dimensions (from fine-grained perception like face/body understanding to high-level reasoning like intention/causal discrimination); (3) High-quality annotations via an automated pipeline (YOLOv11, Qwen2.5-VL, Grounding DINO) plus manual refinement (Gradio interface). The paper conducts extensive experiments on 17 state-of-the-art MLLMs (open/closed-source), revealing critical findings (e.g., stronger scaling effects in Choice/Ranking tasks, training data’s greater impact on grounding than model size, left-right body part discrimination challenges). Contributions include the benchmark itself, a scalable annotation pipeline, and insights guiding MLLM development for human-centric understanding.

**Strengths:**

* Human-MME fills a key gap by covering 8 dimensions that progress from fine-grained perception to abstract reasoning (e.g., causal inference), with 21 question types. This design captures the full complexity of human-centric understanding, unlike existing benchmarks that focus on either coarse perception or single-task reasoning.

- The hybrid automated-manual pipeline balances scalability and accuracy. Automated steps extract fine-grained features (13 bounding box types, 42 facial attributes) using state-of-the-art models, while manual review (cluster de-duplication, instance correction) and identity-neutral processing (removing age/gender from emotion/intention labels) reduce bias and errors—critical for reliable human-centric evaluation.

- Evaluating 17 MLLMs reveals nuanced findings and general trends (scaling benefits Choice/Ranking tasks most). These insights directly guide future MLLM research (e.g., prioritizing grounding data over size for spatial tasks).

**Weaknesses:**

* For the most abstract dimensions (Intention, Causal, Emotion Discrimination), the "ground truth" options are narratives generated by Qwen3. Although these are reviewed by humans, it's unclear if these annotations represent genuine human ground truth or simply a "plausible, model-generated" narrative. Using model-generated text to benchmark the reasoning of other models in such subjective tasks is a potential methodological weakness that merits more discussion.
* The authors need to provide more latest MLLMs to support experimental conclusions, such as Claude-4-Sonnet,  GPT-5, and o3.
* The data and code not be publicly released.

**Questions:**

* The results in Table 2 show that models like Gemini-2.5-Pro and Intern-S1 have very poor grounding (BBox) scores but strong high-level reasoning scores, while GLM-4.5V shows the opposite pattern. Does this suggest a fundamental architectural or training trade-off between fine-grained spatial localization and high-level abstract reasoning, or do you believe this is purely an artifact of their specific training data (as discussed in 3.3)?
* Did you measure difficulty (e.g., inter-model accuracy variance, human annotation time) across dimensions/sub-fields? Are there imbalances (e.g., over-representation of easy Daily Life questions) that might skew model rankings?

I look forward to an active discussion with the authors during the rebuttal phase and will revise my score accordingly.

---

> ### Author Response · Authors · 2025-11-24
>
> We sincerely thank the reviewer for recognition and constructive feedback. We address the raised concerns point-by-point below.
>
> > **Q1. On whether Qwen-generated narratives weaken the validity of ground truth in high-level reasoning tasks**
>
> We appreciate the reviewer's insightful concern and agree that the use of model-generated narratives requires rigorous justification. To evaluate reliability, we conduct a human comparison study on 20% of total images used in Emotion Discrimination. The human annotator produces emotion labels, and we ask another group of annotators (two annotators cross-verify scores) to rate the similarity between human annotations and Qwen's annotations in emotional cues. The average score reaches **8.7/10**, showing strong alignment in core emotional cues. The primary difference lies in Qwen's deliberate choice of expressions that avoid mentioning concrete features, which results from our requirement that the annotator avoid any identity-linked or overly direct cues. This avoidance is deliberate and desirable: it reduces the presence of irrelevant clues and forces the evaluated models to rely on genuine reasoning instead of shortcut signals.
>
> More importantly, the "ground truth" for these tasks is not the textual description itself but the **image-text correspondence**. Each option originates directly from an image using a consistent, controlled pipeline, which means tested MLLMs can't make decisions based on expression style. The correct option comes from the same image while the distractors come from similar but different images. Therefore, the correct choice is intrinsically the one with the strongest correspondence. To avoid the weaknesses of model-generated text serving as absolute ground truth, we intentionally restrict these tasks to **multiple-choice questions** rather than open-ended ones. In open-ended settings, LLM-generated text would indeed be problematic (e.g., bias in LLMs-as-Judge). Our design avoids such issues entirely.
>
> ---
>
> > **Q2. On evaluating more recent models such as Claude-4-Sonnet, GPT-5, o3**
>
> Thank you for this helpful suggestion. We add **GPT-5** and **GPT-5-mini** to the closed-source evaluation, and include **InternVL3.5-241B** among open-source models. These additions strengthen the empirical support for our earlier findings and also yield new insights. The full extended analysis is provided in **Appendix I**.
>
> ---
>
> > **Q3. On releasing data and code**
>
> We appreciate the reviewer's attention to reproducibility. We release partial Human-MME data, additional examples for all 21 QA types (a subset of the original dataset due to limited storage), and the complete code for automatic annotation, QA construction, and the benchmark toolkit. The full dataset will be released with the camera-ready version.
>
> ---
>
> > **Q4. On whether Table 2 suggests a trade-off between grounding (e.g., BBox) and high-level reasoning**
>
> We thank the reviewer for raising this important question. A deeper investigation, detailed in Appendix I, shows that **training data and output alignment**, not architecture or model size, dominate grounding performance.
>
> To examine potential structural trade-offs, we add **InternVL3.5-241B**, a controlled comparison partner to **Intern-S1**. The two models share nearly identical architecture and scale but differ in training data. Under these controlled conditions, InternVL3.5-241B achieves **twice the IoU** of Intern-S1, while maintaining **similar high-level reasoning performance** without directional differences. This indicates that no inherent architectural trade-off exists; rather, previously observed patterns arise from training data composition.
>
> We also analyze why certain closed-source models (e.g., Gemini-2.5-Pro) show strong high-level reasoning but unexpectedly low bounding-box performance. All such models can localize whole faces and bodies accurately, but their performance drops sharply on **fine-grained body parts** (eyebrows, feet, etc.). The drop is much more than comparable open-source model, meaning it is not just because of the task itself becomes harder. It suggests **imbalanced training data**, as public datasets contain abundant full-body and full-face annotations but very limited bounding-box-level annotations for fine-grained parts. Many datasets instead use keypoints or masks. This lack of diverse grounding data limits generalization, revealing a clear direction for future improvements.
>
> These results collectively show that modern MLLMs have **not yet exhausted the visual grounding capacity of their architectures**, and that progress should focus on training data and output alignment. This aligns with very recent work such as *Patch-as-Decodable-Token* (Su et al., 2025), which improves grounding by altering output tokenization and adding a decoder for fine semantic alignment, without modifying the backbone architecture itself.
>
> ---

---

> > ### Author Response · Authors · 2025-11-24
> >
> > > **Q5. On possible difficulty imbalance across dimensions or sub-fields**
> >
> > We appreciate the reviewer's question and include a full difficulty analysis in **Appendix K**. We categorize questions into five difficulty levels based on the number of models that answer correctly, and classify models into five ability groups according to their final scores. We find no evidence of imbalance across difficulty levels, diverse question types, dimensions, and scenarios enable us to evenly cover every difficulty level. Accuracy increases steadily with model capability at all difficulty levels, indicating that each difficulty band provides meaningful discriminative power. No difficulty level disproportionately influences rankings. Overall, the benchmark reliably reflects genuine performance differences.
> >
> > ---
> >
> > We sincerely thank the reviewer for the constructive feedback, insightful questions, and careful reading of our work. The comments substantially help us strengthen the evaluation, analyses, and presentation of Human-MME.

---

> > > ### Comment · Reviewer_rEtP · 2025-11-27
> > > **Response to the author**
> > >
> > > Thanks for the detailed response. I have read the author's rebuttal carefully, along with the other reviews, and most of my concerns have been resolved. Given the paper's scope and comprehensive experiments, I remain mostly positive and will keep my scores.

---

> > > > ### Author Response · Authors · 2025-11-28
> > > >
> > > > Dear Reviewer rEtP,
> > > >
> > > > We sincerely appreciate your careful reading of our paper and your constructive feedback throughout the review process. We are grateful for your recognition of our work and are glad that our rebuttal addressed your concerns. Your comments are really helpful in strengthening the presentation and analyses in Human-MME.
> > > >
> > > >
> > > >
> > > > Best regards,
> > > >
> > > > The Authors

---

### Official Review · Reviewer_jnXs · 2025-10-31

**Soundness:** 2
**Presentation:** 3
**Contribution:** 3
**Rating:** 6
**Confidence:** 3

**Summary:**

This paper presents Human-MME, a large-scale benchmark for evaluating human-centric multimodal large language models (MLLMs). It includes 19,945 image-question pairs across 43 scenarios, covering 8 core human understanding dimensions and 5 question types. Through both automated and manual annotation, the evaluation of 17 major MLLMs reveals clear weaknesses in tasks like left-right hand recognition, emotion analysis, and hallucination suppression.

**Strengths:**

- Pioneers the construction of a systematic, multi-dimensional, end-to-end benchmark for human-centric scene evaluation, offering comprehensive content coverage.

- The benchmark is data-rich and evaluates a wide range of capabilities: from low-level perception (face, body, interaction) to high-level reasoning (emotion, intent, causality, multi-image/multi-object), forming a progressive evaluation hierarchy with rich annotation dimensions (such as left/right hand, emotion, intent, causality).

**Weaknesses:**

- Although the benchmark is extensive in both scale and coverage, its methodology (automated annotation plus manual validation, Q&A-style evaluation) is largely similar to existing benchmarks (e.g., MMBench, MME), lacking methodological novelty.

- The current benchmark focuses mainly on static images, without covering video-based continuous scenarios or multimodal temporal reasoning, which falls short of some human-centric video benchmarks in terms of scenario, action, and intent diversity.

- The analysis of model differences and conclusions is not sufficiently in-depth. While the performance disparities among models are reported, the study lacks a thorough exploration of the underlying causes—for example, the influence of training data, visual encoder, or alignment strategies on the results.

**Questions:**

- Has a quantitative analysis been conducted to prevent data leakage (i.e., training/testing overlap)? If models have potentially seen duplicates such as HICO-DET, how might this affect the evaluation conclusions?

- The remaining issues will be further explored after communicating with the other reviewers.

---

> ### Author Response · Authors · 2025-11-24
>
> # Rebuttal
>
> We sincerely thank the reviewer for the constructive and detailed feedback. Below we address the main concerns regarding methodological novelty, video understanding, depth of analysis, and data leakage.
>
> ---
>
> > **Q1. Methodological Advantages and Novelty of Human-MME**
>
> Although Human-MME adopts a Q&A evaluation format similar to previous benchmarks, its *methodological pipeline*, *data structuring*, and *curation strategy* introduce significant innovations that distinguish it from existing work.
>
> To contextualize the comparison, we summarize major general-purpose multimodal benchmarks below:
>
> ### **Benchmark Comparison**
>
> | Benchmark            | Scale   | Data Source                | Labeling                              | QA Curation | Quality Assurance                             |
> | -------------------- | ------- | -------------------------- | ------------------------------------- | ----------- | --------------------------------------------- |
> | MMBench              | 3.2K    | Public datasets & Internet | No labeling; QAs directly constructed | Human       | Human verification after multiple LLM filters |
> | MME                  | 2.8K    | Public datasets            | No labeling; QAs directly constructed | Human       | Not mentioned                                 |
> | Seed-Bench           | 19K     | Public datasets            | Dataset captions & FM-generated text  | GPT-4       | Human verification                            |
> | HV-MMBench           | 8.7K    | Internet                   | Qwen2.5                               | Qwen2.5     | Human verification after Qwen2.5 filtering    |
> | HumanVBench          | 2.1K    | Internet                   | Video-MLLM generated                  | LLMs        | Human verification                            |
> | Face-Human-Bench     | 2.7K    | Public datasets            | Dataset captions                      | Templates   | Human verification                            |
> | HumaniBench          | 32K     | Internet                   | GPT-4o                                | Templates   | Human verification                            |
> | **Human-MME (Ours)** | **20K** | Public datasets + Internet | **Specialized expert models + LLMs**  | Templates   | **Human filtering + correction**              |
>
> ### **Key Innovations of Human-MME**
>
> ### **(1) Fine-grained specialization enables the use of *reliable expert models***
>
> Focusing exclusively on person-centric images allows us to integrate a large set of expert models (YOLO, DWPose, FaceXFormers, etc.), rather than relying on:
>
> * manual labeling (expensive and limits scale),
> * dataset-provided labels (less flexible; risk data leakage), or
> * single-LLM annotation (introduce biases; hard to test abilities beyond general LLMs').
>
> This makes Human-MME fundamentally different from benchmarks that construct labels solely from captions or LLM outputs.
>
> ### **(2) Structured multi-model extraction → complex, compositional QA creation**
>
> Our pipeline extracts highly structured human attributes from multiple specialized models, forming a comprehensive feature graph for each person in the image.
> This enables question construction that requires **multi-step reasoning and feature composition**, e.g.:
>
> > *“What is the age group of the person who is wearing the beige jumpsuit?”*
>
> Template-based QA generation becomes feasible without relying on large language models, avoiding the circularity and bias inherent in LLM-generated QA.
>
> ### **(3) Cross-image redundancy filtering to improve annotation reliability**
>
> Many images come from similar sources with visually similar or near-duplicate frames.
> We systematically compare and filter labels across these images to:
>
> * identify the most reliable annotation,
> * reduce human effort,
> * increase diversity while improving annotation quality.
>
> This multi-source aggregation strategy is not present in prior benchmarks.
>
> ---

---

> > ### Author Response · Authors · 2025-11-24
> >
> > > **Q2. On Video vs Image Benchmarks**
> >
> > The reviewer notes that Human-MME focuses on static images. We agree and clarify:
> >
> > * **Image and video benchmarks assess different abilities.**
> >   For instance, *causal discrimination* in images requires predicting unobserved events using commonsense (e.g., a girl running toward a hat in the sky suggests the wind has just blown it from her hand). In videos, causality typically refers to explaining events that are directly observable through motion cues (e.g., seeing the hat fly off due to the wind). Video understanding relies on meticulous observation of dynamic phenomena, whereas image-based causality demands stronger inference with limited clues. We clarify that both tasks are uniquely valuable but belong to different domains, and our benchmark complements this distinction.
> > * **Our QA curation pipeline is image-specific**, and does not yet support temporal modeling or dynamic sequence annotation.
> >
> > Thus, extending Human-MME to video is indeed important future work, but remains **orthogonal** to our contributions on fine-grained human-centric image understanding.
> >
> > ---
> >
> > > **Q3. Additional Model Analysis Added in the Revision**
> >
> > With additional benchmark results from GPT-5, GPT-5-mini, and InternVL3.5-241B, we have more data to analysis and provide insight. We expand Appendix E and Appendix I to provide deeper analysis of 20 major MLLMs, including architecture details, training alignment, grounding strategies, and failure patterns.
> >
> > ### **Deeper Analysis of Key Findings** (full list of 7 findings in Appendix I):
> >
> > 1. **Training data dominates performance** on human-related grounding, more than architecture or model size. We find more evidence with new and existing results.
> > 2. **CoT alignment in Qwen2.5-VL-72B** improves hallucination resistance but sometimes makes the model overly conservative.
> > 3. **GPT-5's safety alignment** significantly weakens instruction-following behaviors in human-related questions.
> > 4. All four tested closed-source models achieve **good coarse grounding but rapidly degrade on fine-grained grounding**, suggesting unbalanced training data.
> >
> > ---
> >
> > > **Q4. Data Leakage Analysis**
> >
> > Our benchmark labels come from our own annotation pipeline, so leakage risk mainly concerns **images**, not labels, which greatly reduce the risk and effect of data leakage.
> > We conduct both **membership inference attacks** and **clean-subset re-evaluation** to ensure robustness.
> >
> > ### **(1) Membership Inference Attack (MIA)**
> >
> > We apply **VL-MIA** (Li et al., NeurIPS 2024) to test whether Human-MME images resemble leaked or non-leaked distributions.
> >
> > We compute Rényi proposed in the paper for our images and compare them using the KS statistic:
> >
> > * KS w.r.t. leaked set = **0.14 (p = 2.1e-5)**
> > * KS w.r.t. non-leaked set = **0.06 (p = 0.21)**
> >
> > **Human-MME images are statistically closer to non-leaked data**, indicating low leakage likelihood.
> >
> > ### **(2) Clean Subset Evaluation (Removing All HICO-DET-related QAs)**
> >
> > Since HICO-DET is older and more likely to be in training data, we assume it has data leakage and remove all questions derived from these images and recompute scores to see whether there is a impact .
> >
> > | Model Name      | Original | Clean | Δ    |
> > | --------------- | -------- | ----- | ---- |
> > | GLM-4.5V        | 76.0     | 77.0  | +1.0 |
> > | Qwen2.5 VL-72B  | 72.8     | 73.3  | +0.5 |
> > | Qwen2.5 VL-32B  | 69.1     | 69.7  | +0.6 |
> > | GLM-4.1V-9B     | 69.1     | 69.5  | +0.5 |
> > | Gemini-2.5-Pro  | 68.9     | 69.2  | +0.3 |
> > | InternVL3-78B   | 68.9     | 68.5  | −0.3 |
> > | Intern-S1       | 68.1     | 68.4  | +0.2 |
> > | InternVL3.5-38B | 67.6     | 66.9  | −0.8 |
> > | MiniCPM-V-4.5   | 63.0     | 63.2  | +0.3 |
> > | Qwen2.5 VL-7B   | 62.8     | 63.1  | +0.3 |
> > | Gemma3-27B      | 60.1     | 59.5  | −0.7 |
> > | LLaVA-NeXT-72B  | 59.3     | 58.6  | −0.7 |
> > | GPT-4o          | 59.0     | 57.2  | −1.8 |
> > | Aya-vision-32B  | 57.7     | 56.8  | −0.9 |
> > | Kimi-VL-A3B     | 52.6     | 53.0  | +0.5 |
> > | Llama-4-Scout   | 48.0     | 46.3  | −1.8 |
> > | Phi-4           | 42.8     | 42.7  | −0.2 |
> >
> > **Model rankings remain almost unchanged; top-10 model shifts are within ±1.0.**
> > This confirms that even if some leakage exists, its effect on evaluation is negligible, which is consistent with claims in MME (Fu et al., 2025).
> >
> > ---
> >
> > ## **References**
> >
> > * Li et al., *Membership Inference Attacks against Large Vision-Language Models*, NeurIPS 2024.
> > * Fu et al., *MME: A Comprehensive Evaluation Benchmark for Multimodal LLMs*, NeurIPS 2025.
> >
> > ---
> >
> > We appreciate the reviewers' feedback and hope this clarifies the novelty and rigor of our work. Revisions based on your feedback further strengthen the clarity and completeness of our work. Thank you again for your valuable feedback.

---

### Official Review · Reviewer_cQVR · 2025-11-01

**Soundness:** 2
**Presentation:** 2
**Contribution:** 3
**Rating:** 4
**Confidence:** 4

**Summary:**

The paper proposes Human-MME, a large-scale benchmark for evaluating multimodal large language models on human-centric scene understanding, covering eight progressive dimensions from fine-grained perception to high-level reasoning, with 19945 samples, an automated–manual annotation pipeline, and evaluations on 17 models.

**Strengths:**

1. The paper addresses the underexplored problem of human-centric multimodal understanding and provides a clear motivation for its importance.
2. It provides a holistic eight-dimension evaluation that spans fine-grained perception to high-level reasoning, with 19,945 real-world image–question pairs covering 43 diverse scenarios across four human activity domains.
3. The hybrid automated and expert-reviewed annotation pipeline combines scalability with high annotation quality and reduced bias.

**Weaknesses:**

1. Proprietary models such as GPT-4o perform poorly on face-related or human-identity tasks, which may stem from privacy or policy restrictions, but the paper does not discuss this possible factor.
2. The evaluation could be improved by including results from newer proprietary models to better reflect the current state of the field.
3. Lacks deeper error analysis, especially for the best-performing models, to identify detailed failure patterns across tasks and provide clearer insights for future model development.

**Questions:**

1. Could the authors clarify whether the poor performance of proprietary models such as GPT-4o on face-related or human-identity tasks is due to policy or privacy restrictions, or inherent model limitations?
2. Would it be possible for the authors to include or discuss results from newer proprietary models to better reflect the current performance landscape?
3. Could the authors provide a more detailed error analysis for the best-performing models, like highlighting representative failure cases across different task types to better understand current limitations?

---

> ### Author Response · Authors · 2025-11-24
>
> # Rebuttal
>
> We sincerely thank the reviewer for the constructive feedback and for acknowledging the motivation, data quality, and comprehensive design of *Human-MME*. We address the raised concerns point-by-point below.
>
> ---
>
> > **Q1. On the poor performance of proprietary models on face-related or human-identity tasks**
>
> The reviewer suggests that the inferior performance of proprietary models (e.g., GPT-4o) may be due to policy or privacy restrictions. We carefully examine this possibility.
>
> **(1) GPT-4o does not show evidence of policy-driven refusals.**
> Across all evaluation dimensions, GPT-4o's answers do not contain refusal-related keywords (e.g., "sorry", "policy", "privacy"). This indicates that GPT-4o's errors largely stem from inherent model limitations, rather than from enforced safety constraints.
>
> **(2) GPT-5 and GPT-5-mini do exhibit policy-driven refusals.**
> We add evaluations of GPT-5 and GPT-5-mini, both of which contain frequent refusals when questions involve identity-related information such as gender, race, and facial attributes. We identify these refusals by searching for keywords associated with safety alignment ("policy", "privacy", "not appropriate", etc.).
>
> Below is the distribution (in %) of policy-triggered refusals across the eight dimensions:
>
> | Model          | FU      | BU  | HU  | MIU     | MPR     | ID  | CD      | ED      | Avg.    |
> | -------------- | ------- | --- | --- | ------- | ------- | --- | ------- | ------- | ------- |
> | Gemini-2.5-Pro | 0.0     | 0.0 | 0.0 | 0.0     | 0.0     | 0.0 | 0.0     | 0.0     | 0.0     |
> | GPT-4o         | 0.0     | 0.0 | 0.0 | 0.0     | 0.0     | 0.0 | 0.0     | 0.0     | 0.0     |
> | **GPT-5**      | **2.1** | 0.0 | 0.0 | **0.3** | **1.7** | 0.0 | 0.0     | **7.3** | **1.4** |
> | **GPT-5-mini** | **3.8** | 0.0 | 0.0 | **0.6** | **3.2** | 0.0 | **0.2** | **6.1** | **1.7** |
>
> More detailed discussion is provided in **Appendix I** (including additional tables referenced as Table 4 and 5 in section 3.3).
>
> ---
>
> > **Q2. Inclusion of newer proprietary models**
>
> Following the reviewer's suggestion, we add results for the latest GPT-5 and GPT-5-mini models. Detailed results are added to Table 2 and Table 8.
>
> | Model          | FU   | BU   | HU   | MIU  | MPR  | ID   | CD   | ED   | Avg  |
> | -------------- | ---- | ---- | ---- | ---- | ---- | ---- | ---- | ---- | ---- |
> | Gemini-2.5-Pro | 42.4 | 66.5 | 70.0 | 83.6 | 48.6 | 79.4 | 86.1 | 64.5 | 67.6 |
> | GPT-4o         | 28.8 | 58.8 | 59.8 | 74.7 | 34.4 | 79.2 | 76.2 | 52.7 | 58.1 |
> | GPT-5          | 34.4 | 67.8 | 71.1 | 75.8 | 43.1 | 82.3 | 89.2 | 42.6 | 63.3 |
> | GPT-5-mini     | 30.6 | 66.3 | 67.4 | 76.4 | 41.3 | 79.4 | 81.7 | 39.9 | 60.4 |
>
> With newly obtained data, we expand the error analysis of all proprietary models and summarize the major findings as follows:
>
> ### **(1) Proprietary models handle global regions well but collapse on fine-grained grounding.**
>
> Across GPT-5, GPT-5-mini, Gemini-2.5-Pro, and GPT-4o, we observe a consistent pattern:
>
> * Whole-body / whole-face IoU is reasonable, indicating coarse localization ability.
> * Fine-grained body-part and facial-part IoU is dramatically worse, often only 5%-20% of the open-source baseline.
>
> These results are quantitatively shown in Table 6 in Section 3.3 and further analyzed in Appendix I, including the detailed IoU comparisons. This indicates that training data imbalance, where whole-body/whole-face annotations are abundant but part-level annotations are rare, is a likely cause. This gap may stem from the slower iteration pace of closed-source models, coupled with the fact that the requirements for fine-grained visual alignment in semantic tasks tend to favor scenarios where closed-source models is not a good choice. Common scenarios for fine-grained visual alignment in semantic tasks include large-scale data annotation and fine-tuning for downstream tasks, both of which are more conducive to open-source models. Consequently, the latest open-source models have also been optimized for this task accordingly.
>
> ### **(2) GPT-5 series suffers from excessive safety filtering.**
>
> In GPT-5 and GPT-5-mini:
>
> Safety mechanisms are frequently triggered by tasks requiring inference of appearance-related attributes. The refusal message often replaces the expected structured answer entirely. This greatly affects certain dimensions (e.g., Face Understanding, Emotion Discrimination).
>
> ### **(3) GPT-4o underperforms due to being relatively old (mid-2024).**
>
> Multiple benchmarks (e.g., MathVista, MME, MME-Realworld) also show a similar degradation pattern.
> Therefore, GPT-4o’s weaker performance is expected and not due to policy constraints.
>
> All these analyses are included in the newly added expanded discussion in **Appendix I**.
>
> ---

---

> ### Author Response · Authors · 2025-11-24
>
> > **Q3. Detailed error analysis for the best-performing models**
>
> Thank you for the suggestion. We add a more focused case study for the strongest models (GLM-4.5V and Gemini-2.5-Pro) in **Appendix J**, and summarize the key insights here.
>
> **(1) Premature decision-making in GLM-4.5V.**
> GLM-4.5V often commits to an answer before fully evaluating all options, especially in fine-grained facial-attribute tasks. This "answer-first, reasoning-after" behavior leads to systematic misclassification.
>
> **(2) Difficulty with fine-grained clothing categories.**
> GLM-4.5V is easily distracted by visually salient but category-incorrect garments (e.g., mistaking a suite jacket for described fluffy jacket), and struggles to distinguish similar categories in multi-person comparisons.
>
> **(3) Weak inference from bounding-box coordinates.**
> Although GLM-4.5V predicts boxes well, it frequently misidentifies what lies inside a given box, relying on rough positional heuristics rather than precise spatial reasoning.
>
> **(4) Orientation-related errors in Gemini-2.5-Pro.**
> Gemini-2.5-Pro occasionally confuses left–right facial orientation, affecting tasks that depend on directional cues and attribute counting.
>
> ## **References**
>
> * Lu et al., 2024. MathVista: Evaluating math reasoning in visual contexts with GPT-4V, Bard, and other multimodal models. ICLR 2024.
> * Fu et al., 2024. MME: A comprehensive evaluation benchmark for multimodal large language models. NeurIPS 2025.
> * Cai et al., 2025. HV-MMBench: Benchmarking MLLMs for human-centric video understanding. arXiv:2507.04909.
>
> ---
>
> Revisions based on your feedback further strengthen the clarity and completeness of our work. Thank you again for your valuable feedback.

---

> ### Comment · Reviewer_cQVR · 2025-11-24
>
> Thank you for your detailed response. Most of my concerns have been addressed, and I will raise my score.

---

> > ### Author Response · Authors · 2025-11-25
> >
> > Dear Reviewer cQVR,
> >
> > Thank you for your positive feedback! We are pleased to hear that our rebuttal has successfully addressed your concerns. We sincerely appreciate the time and effort you have taken to provide detailed reviews, which have significantly helped to improve the quality of our manuscript.
> >
> > Best regards,
> >
> > The Authors

---

### Author Response · Authors · 2025-11-24
**General Responses**

We sincerely appreciate the constructive and valuable feedback from all reviewers. After carefully considering their comments, we revise our submission. Below we summarize the additional experiments, analyses, and discussions included in the updated version for common concerns of reviewers.

1. Expanded evaluation results (**Table 2**). We extend our evaluation to include GPT-5, GPT-5-mini, and InternVL3.5-241B. The total number of evaluated models increases from 17 to 20.

2. More comprehensive comparison with existing benchmarks (**Table 1**, **Table 7**). We broaden the comparison to the methodological level, providing a clearer illustration of the advantages offered by our benchmark.

3. Analysis of underwhelming closed-source model performance (**Section 3.3**, **Table 4**, **Table 5**, **Table 6**, **Appendix I**). Our study reveals that the latest closed-source GPT-5 exhibits degraded performance in facial and emotion analysis due to its stricter safety alignment. GPT-4o, although not affected by safety alignment, is limited by its outdated capability level. All evaluated closed-source models, including Gemini-2.5-Pro, show noticeable disadvantages in fine-grained tasks despite possessing vision grounding abilities.

4. Deeper summary of open-source model architectures and training data (**Appendix E**, **Table 10**, **Appendix I**). We provide a detailed analysis of the architectures and training data of 16 open-source models. Our findings highlight that training data and alignment format, rather than model size, vision encoder, or language backbone, play the most decisive role in fine-grained grounding performance.

5. Case study of misbehavior in high-performing models (**Appendix J**). Our qualitative analysis uncovers several systematic issues. GLM-4.5V sometimes exhibits answer-first reasoning-after behavior. Models show misunderstandings regarding certain clothing categories. They demonstrate weak ability in inferring the correct object from relative coordinate values, which is the dual problem of bounding box prediction. Gemini-2.5-Pro also suffers from left-right confusion, similar to open-source models.

6. Analysis of difficulty distribution in Human-MME (**Appendix K**, **Figure 10**). We examine the difficulty distribution of the benchmark and show that Human-MME has a diverse and well-balanced set of questions, with good discriminative power across all difficulty levels.

7. Demonstration of dataset diversity (**Figure 8**). We additionally provide representative samples to illustrate the diversity of populations, scenes, and cultural contexts covered by the benchmark images.

---

### Author Response · Authors · 2025-12-02

We would like to express our sincere gratitude to the reviewers, area chairs, senior area chairs, and program chairs for their swift and concerted efforts in addressing this unexpected information leakage issue.

Our initial responses comprehensively clarified and resolved the questions raised by all reviewers. Most notably, **Reviewer cQVR** explicitly acknowledged that most concerns were addressed and confirmed the decision to **raise the rating to 6** on **24 Nov 2025**. This confirmation was finalized before the large-scale leakage of information, ensuring that the positive assessment was based solely on the quality of our manuscript and rebuttal.

In our general response, we have already summarized the extensive additional experiments conducted to address these concerns, including expanded evaluations on the latest models (GPT-5, GPT-5-mini, InternVL3.5-241B), training-test overlap analysis, difficulty analysis, and in-depth error case study for high-performing models. These substantive improvements have enhanced the rigor of our submission and were well-received by reviewers cQVR and rEtP.

We sincerely hope that our meticulous response to reviewer comments, coupled with the positive feedback confirmed prior to the incident, will facilitate a fair and comprehensive assessment of our work.

---

### Meta-Review · Area_Chair_Pnjs · 2026-01-09

**Summary:**

The paper proposes Human-MME, a large-scale benchmark for evaluating multimodal large language models on human-centric scene understanding. The key features of Human-MME include: (1) diverse human scenes (4 primary domains, 15 secondary domains, and 43 subfields) and 19,945 real-world image–question pairs; (2) progressive evaluation across eight dimensions (from fine-grained perception, such as face/body understanding, to high-level reasoning, such as intention and causal discrimination); and (3) high-quality annotations produced via an automated pipeline with manual refinement.
Before the rebuttal, the paper received mixed ratings: two reviewers rated it “4 (marginally below threshold),” one reviewer rated it “6 (marginally above threshold),” and one reviewer rated it “8 (good paper/poster).” The authors provided a detailed rebuttal that addressed most of the reviewers’ concerns.

**Reviewer Concerns:**

Most issues raised by the reviewers are addressed in the rebuttal, and there are no major outstanding issues.

Specifically, the authors introduced the below key changes during the rebuttal:

(1) Extended the evaluation to include GPT-5, GPT-5-mini, and InternVL3.5-241B.
(2) Provided additional explanations and comparison with existing benchmarks, and provided additional analysis of experimental results.
(3) Provided additional analysis on the difficulty distribution and diversity of the benchmark.

Outstanding:

No major outstanding issues based on the reviewer comments. However,  the AC expect the below points to be addressed in the final draft.
(i)  The authors have promised to open-source the dataset and annotations. However, as the proposed dataset contains facial images, it is important to confirm that the authors have permission to redistribute such images and that it is not violating privacy policies, especially for the 8k images sourced by the authors from free media sites. Also, even for public datasets, it is important to confirm the license terms to ensure permission to redistribute or repurpose the images.
(ii) Additionally, the manual annotation details are not provided in the draft. How many human annotators are involved in the process? How are they rewarded for the annotation task?

**Reviewer Scores:**

As the authors provided a very detailed rebuttal that addressed most of the concerns raised by the reviewers, I believe that reviewers cQVR and ZsgC would have increased their ratings from “4 (marginally below threshold)” to “6 (marginally above threshold),” while reviewers rEtP and jnXs would have maintained their positive “accept” and “marginally above threshold” ratings.

---

### Decision · Program_Chairs · 2026-01-26

Accept (Poster)